# Symbolic Density Estimation:
# A Decompositional Approach

## Abstract

We introduce AI-Kolmogorov, a novel framework for Symbolic Density Estimation (SymDE). Symbolic regression (SR) has been effectively used to produce interpretable models in standard regression settings but its applicability to density estimation tasks has largely been unexplored. To address the SymDE task we introduce a multi-stage pipeline: (i) problem decomposition through clustering and/or probabilistic graphical model structure learning; (ii) nonparametric density estimation; (iii) support estimation; and finally (iv) SR on the density estimate. We demonstrate the efficacy of AI-Kolmogorov on synthetic mixture models, multivariate normal distributions, and three exotic distributions, two of which are motivated by applications in high-energy physics. We show that AI-Kolmogorov can discover underlying distributions or otherwise provide valuable insight into the mathematical expressions describing them.

## 1. Introduction

Data-driven approaches are widely used to model complex nonlinear relationships between predictive features and target variables. Popular approaches tend to be black-box and do not easily yield insight into underlying patterns in the data. Symbolic Regression (SR) is a family of methods that offer an interpretable alternative by searching the space of mathematical expressions and parameters to produce models that reveal useful insights into the structure underlying the data. While SR has seen success in supervised learning tasks such as regression and differential equation discovery, extending SR to the **unsupervised task of density estimation** introduces unique challenges and remains largely unexplored.

[1]Anonymous Institution, Anonymous City, Anonymous Region, Anonymous Country. Correspondence to: Anonymous Author <anon.email@domain.com>.

Preliminary work. Under review by the International Conference on Machine Learning (ICML). Do not distribute.

Classical methods for density estimation typically fall into one of two categories; parametric density estimation assumes a model for the distribution and finds the best-fitting parameters, and nonparametric density estimation avoids model assumptions but sacrifices interpretability. This work addresses the problem of Symbolic Density Estimation (SymDE) using symbolic regression to discover mathematical expressions that accurately describe continuous probability distributions. Searching the space of closed-form expressions avoids the restrictive model assumptions of parametric density estimation while retaining interpretability unlike nonparametric density estimation.

The SymDE problem faces three primary challenges: (i) ensuring validity constraints on probability distributions, namely non-negativity and normalization over the support of the distribution; (ii) the curse of dimensionality; and (iii) the difficulty of discovering complex symbolic expressions from a vast search space.

**Related Work** SR is an established area of research with several promising deep learning based approaches emerging over recent years. However, literature on the application of SR to discover probability distributions from raw samples is scarce. Aside from seminal works on SR, few works that address the SymDE are mentioned here.

*Symbolic Regression:* SR is classically approached with genetic programming (GP) (Koza, 1992), with PySR (Cranmer, 2023) currently among the state-of-the-art tools. These algorithms employ genetic operations (crossover, mutation) to evolve expressions over successive generations, evaluating them using some measure of fitness. The methods presented in this paper use PySR as the engine for SymDE, although another popular SR engine may also be used.

*Deep Learning-Based SR:* Recent advances that apply deep learning methods to discover descriptive mathematical expressions directly from data include Deep Symbolic Regression (DSR) (Petersen et al., 2021) which uses reinforcement learning (RL). Neural-Guided Genetic Programming (NGGP) (Mundhenk et al., 2021) combines RL with GP. Transformer-based techniques include NeSymReS (Biggio et al., 2021), SymbolicGPT (Valipour et al., 2021), and E2ESR (Kamienny et al., 2022). Finally, LASR (Grayeli

et al., 2024) enhances GP by integrating learned concept libraries guided by LLMs, introducing a mechanism that brings the technique closer to the human scientific discovery process.

*Decomposition-Based Methods:* Other methods like AI Feynman (Udrescu & Tegmark, 2020) and AI Feynman 2.0 (Udrescu et al., 2020), integrate neural networks with recursive decomposition strategies for equation discovery, aiming to reduce the combinatorial explosion of the symbolic search space. Symbolic Regression using Control Variables (Jiang & Xue, 2023; Chu et al., 2024) also attempts to address this issue by decomposing multi-variable SR into single-variable problems. These methods directly inspire our technique of decomposing high-dimensional distributions into simpler components.

*Symbolic Density Estimation:* Maximum-Entropy based stochastic and symbolic density estimation (MESSY) (Tohme et al., 2024) is a symbolic density estimation framework that recovers analytical forms of probability density functions from sample data. It constructs a gradient-based drift–diffusion process that connects observed samples to a maximum entropy ansatz. MESSY uses symbolic regression to explore candidate basis functions for the exponent in the maximum entropy model. This approach yields tractable symbolic descriptions of densities with low bias and a cost that is linear in the number of samples and quadratic in the number of basis functions. While the maximum entropy structure ensures valid probability densities and enables recovery of multimodal distributions through refinement, it does constrain the class of expressible densities compared with fully general functional forms.

Morales-Alvarado 2025 apply the LLM-SR (Large Language Model for Symbolic Regression) framework from Shojaee et al. 2025 to benchmark problems of equation recovery in lepton angular distributions and functional forms for angular coefficients in electroweak boson production. LLM-SR treats equations as programs with mathematical operators and combines LMMs' scientific prior knowledge with evolutionary search over equation programs. It iteratively proposed new equation skeleton hypotheses which are subsequently optimized against to find the best parameters. While it takes advantage of available scientific domain knowledge, LLM-SR is limited to the available training data which may lead to biases or gaps.

**Our Contributions** We introduce AI-Kolmogorov, a framework for solving the SymDE problem and bridge the gap between nonparametric and parametric density estimation. While previous approaches address partial aspects of this problem, they do not search over truly general symbolic functional forms for distributions in higher dimensional settings. Symbolic regression tools cannot directly solve

SymDE because the training dataset typically does not contain density labels. We explore evolutionary algorithms such as in PySR (Cranmer, 2023) which could use log-likelihood scores but find that the resultant expressions may not adhere to the non-negativity and normalization constraints that every valid distribution must satisfy.

We conduct experiments that demonstrate the efficacy of AI-Kolmogorov and showcase its results on various datasets. Our evaluation includes synthetic datasets such as multivariate normal distributions and mixture models, and other problems using non-standard distributions. We present ablations studies that evaluate the utility of the decomposition techniques proposed. The results show that AI-Kolmogorov provides useful information about the structure of the underlying distributions, producing interpretable symbolic models with low residual error and effectively trading off model complexity and accuracy, demonstrating the framework's ability to aid the discovery of interpretable mathematical expressions that describe probability distributions. [1]

## 2. Background

**Symbolic Regression** Given a dataset $\mathcal{D} = \{(\mathbf{x}_i, y_i)\}_{i=1}^N$, the goal is to identify a function $f$ such that the target variable $y_i \approx f(\mathbf{x}_i) \quad \forall i \in \{1, 2, ..., N\}$. SR discovers mathematical expressions that predict the target from the feature variables while balancing model accuracy and complexity. The ideal models are accurate yet parsimonious. Unlike traditional regression that fits parameters for a fixed model class, SR instead explores a vast space of potential mathematical expressions to find the most suitable representation. The search space of expressions grows exponentially with the number of permitted operators and variables. Candidate models are constructed as expression trees from operators and variables, adhering to a context-free grammar. SR has been shown to be NP-hard (Virgolin & Pissis, 2022), but despite the computational challenges, SR offers the possibility of interpretability which is central to scientific discovery.

**Probability Distributions and Density Estimation** A probability distribution assigns probabilities to outcomes of a random variable. For continuous variables, the probability density function (PDF) $f_{\mathbf{X}}(\mathbf{x})$ satisfies $f_{\mathbf{X}}(\mathbf{x}) \geq 0$ (non-negativity constraint) and $\int f_{\mathbf{X}}(\mathbf{x}) \, d\mathbf{x} = 1$ (normalization constraint).

Density estimation approximates the probability distribution from samples. Given i.i.d. samples $\mathbf{X}_1, \mathbf{X}_2, ..., \mathbf{X}_n$ from unknown distribution $f_{\mathbf{X}}(\mathbf{x})$, the goal is find an estimate $\hat{f}_{\mathbf{X}}(\mathbf{x})$. Kernel Density Estimation (KDE) is a widely used calssic nonparametric technique where each data point

---

[1]https://anonymous.4open.science/r/SymbolicDensityEstimation-31D4

contributes a localized kernel function,

$$\hat{f}_n(\mathbf{x}) = \frac{1}{nh^d} \sum_{i=1}^{n} K\left(\frac{\mathbf{x} - \mathbf{X}_i}{h}\right),$$

where $h$ is the bandwidth (a smoothing hyper-parameter) and $K$ is the kernel function. While adequate for simple distributions, it's performance suffers in higher dimensions or for distributions that have sharp discontinuities at the boundaries of the support.

Normalizing Flows (NFs) have recently emerged as a powerful alternative for nonparametric density estimation, particularly in high-dimensional spaces. Unlike KDE, which constructs a density by summing local kernels, NFs learns an expressive density by transforming a simple base distribution $f_{\mathbf{Z}}(\mathbf{z})$ (e.g., a standard multivariate Gaussian) into a complex target distribution via a sequence of invertible, differentiable mappings $\mathbf{x} = g(\mathbf{z})$. By parameterizing $g$ with deep neural networks that ensure tractability of the Jacobian determinant, NFs can model highly intricate distributions while allowing for exact likelihood evaluation and efficient sampling.

In Symbolic Density Estimation (SymDE), the challenge is not only to evaluate the density function at any given query points, but to obtain a symbolic description of the distribution. SymDE goes beyond the familiar families of distributions used in parametric density estimation and the black-box nature of nonparametric density estimation to discover descriptive mathematical expressions for exotic distributions.

## 3. Method

### 3.1. The AI-Kolmogorov framework

The framework is illustrated in Figure 1. To discover the underlying continuous probability density function, we employ optional preprocessing stages to decompose the problem. Then non-parameteric density estimation produces a black-box model of the distribution that can be evaluated at any query point to provide a density estimate. This is followed by symbolic regression to obtain candidate symbolic expressions for the underlying probability density function. The nonparametric density estimate acts as a surrogate target for symbolic regression imposing non-negativity and normalization as soft constraints via the mean squared error (MSE) loss computed over a regular grid of evaluation points in the distribution's support.

**Clustering** Clustering is useful for multi-modal data where the modes are well separated. The raw samples are segmented into unimodal clusters before calling symbolic regression on each distinct cluster. Once expressions for each cluster are identified, they can be combined additively.

The model complexity for each cluster individually may be simpler than that for the full distribution, improving the odds of SR finding simpler expressions that can be combined afterwards. An example of an algorithm that can be used for this decomposition is DBSCAN (Wang et al., 2019), a density-based clustering method that automatically infers the number of clusters without parametric assumptions on the shape of the clusters.

**Structure Learning** If clustering enables additive decomposition,then structure learning enables multiplicative decomposition. Structure learning entails identifying the dependence or independence between covariates or equivalently, determining the underlying Probabilistic Graphical Model (PGM). If independent subsets of covariates are detected the computational cost of SymDE on high-dimensional distributions can be dramatically reduced. Each independent subset of covariates appear as a separate connected component in the PGM. SR can be performed on the subset of covariates within each connected component separately, reducing the dimensionality of the problem to be tackled. For example, in the case of a 4-dimensional distribution where two pairs of covariates are independent, the joint distribution can be factorized as a product of two 2-dimensional distributions. We use the constrained-based learning PC algorithm (Neapolitan, 2003) (refer to Appendix B) for structure-learning in continuous data. This makes use of statistical tests to construct probabilistic graphical models and identify the connected components to significantly reduce computational cost for SR.

**Nonparametric Density Estimation** Nonparametric Density Estimation serves as the bridge between raw data and symbolic regression, producing a black-box model of the unknown distribution. With Kernel Density Estimation (KDE), each data point contributes a localized "bump" and the overall estimate is obtained by summing these contributions over the support. We employ Gaussian kernels and select the bandwidth using cross-validation.

In cases where KDE provides inadequate density estimates we use Neural Spline Flows (NSF) (Durkan et al., 2019), a powerful class of normalizing flows. While simpler NFs often rely on affine mappings, NSF utilizes monotonic rational-quadratic splines to transform a simple base distribution into a complex target distribution. By using a neural network to parameterize the locations and derivatives of the spline knots, the model can capture intricate, multi-modal features that simpler flows miss. This combination of high expressivity and numerical stability makes NSF a state-of-the-art choice for modeling high-dimensional data where traditional non-parametric methods like KDE may struggle with the curse of dimensionality.

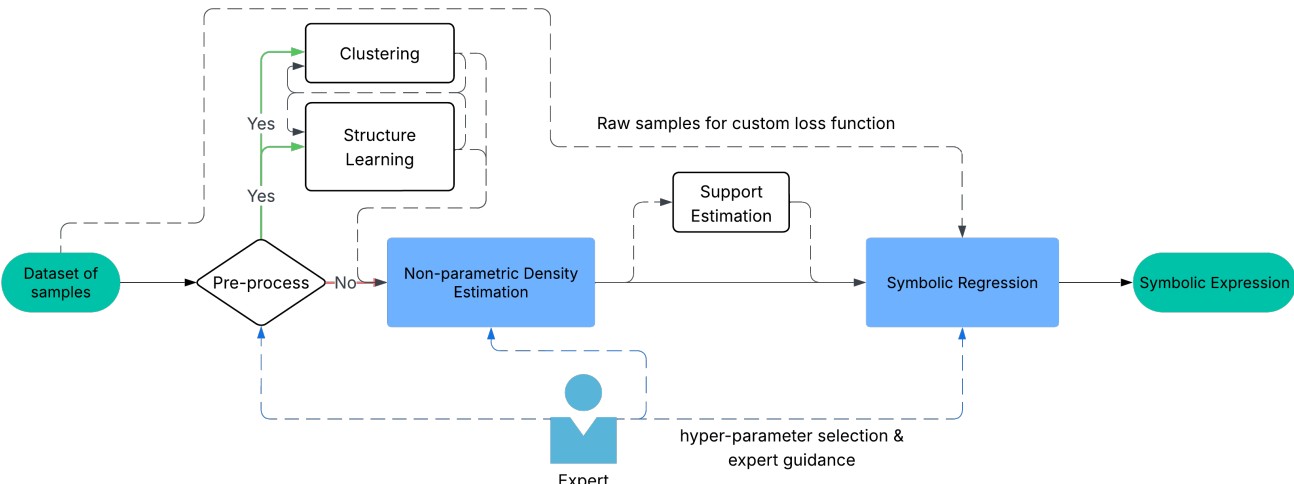

*Figure 1.* The AI-Kolmogorov pipeline: decomposition (clustering/structure learning), density estimation, support estimation, symbolic regression, and warm-start refinement. Optional workflows are indicated by dashed arrows.

**Support Estimation** Support estimation identifies the subset of the input space $\mathcal{X} \subseteq \mathbb{R}^d$ where the density function $f_{\mathbf{X}}(\mathbf{x})$ is strictly positive, preventing the algorithm from searching for expressions that attempt to fit discontinuites at the boundaries of a distribution. We employ level-set thresholding on the non-parametric density estimate: $\widehat{\mathcal{S}}_\tau = \{\mathbf{x} \in \mathbb{R}^d | \hat{f}_n(\mathbf{x}) \geq \tau\}$ where $\tau > 0$ is a small threshold. Alternatively for distributions with convex supports, the convex hull of the samples may be used to estimate the support. To correct any bias of the non-parametric density estimates at boundaries with sharp discontinuities, we employ the reflection trick if the boundaries are straight lines or otherwise apply a shrinking factor to the estimated support to avoid trim regions near the boundary where we know the density estimate is inaccurate. Refer to Appendix B.1 for more details.

**Symbolic Regression** The density estimate from the non-parametric model is used as the target variable in SR. The MSE loss with respect to the density estimate serves as a surrogate loss function for density estimation. We make use of PySR as the computational engine for Symbolic Regression, which employs multi-population evolutionary algorithms to search the space of mathematical expressions. Each candidate expression is evaluated using a fitness function that balances the MSE loss against model complexity as measured by the size of the expression tree. This multi-objective optimization approach favors the discovery of expressions that are both accurate and parsimonious. The set of operators for each problem may be selected based on expert knowledge. For all examples provided in this work, the set of permitted operators is a superset of the operators that appear in the ground truth expression (except for the results in Section 4.4). The output of SR is pareto front of expressions trading off accuracy against complexity allowing users to

examine a zoo of models. Simpler models capture coarse structure and more complex models are able to capture even fine grained variation in the data.

## 4. Results

We evaluate AI-Kolmogorov to demonstrate its merits and weaknesses on various datasets. For all datasets, we assume sample sizes are sufficiently large.

### 4.1. Additive and Multiplicative Decomposition

Clustering and structure learning provide two complementary strategies for decomposing complex distributions into simpler sub-problems. Clustering assumes an additive factorization and reduces the complexity of the symbolic expressions to be discovered per component, while structure learning assumes multiplicative factorization and reduces both target expression complexity and the dimensionality per component.

To demonstrate clustering, we consider $\mathbf{X} \in \mathbb{R}^2$ drawn from a two-component Gaussian mixture:

$$f_{\mathbf{X}}(\mathbf{x}) = \tfrac{1}{2}\mathcal{N}(\mathbf{x} \mid \boldsymbol{\mu_1}, \boldsymbol{\Sigma}) + \tfrac{1}{2}\mathcal{N}(\mathbf{x} \mid \boldsymbol{\mu_2}, \boldsymbol{\Sigma}),$$

with

$$\boldsymbol{\mu_1} = \begin{pmatrix} -4.0 \\ 4.0 \end{pmatrix}, \quad \boldsymbol{\mu_2} = \begin{pmatrix} 4.0 \\ -4.0 \end{pmatrix}, \quad \boldsymbol{\Sigma} = \begin{pmatrix} 1 & 0.8 \\ 0.8 & 1 \end{pmatrix}.$$

This distribution can be decomposed the sum of two Gaussian components. We denote $\mathbf{x} = (x_1, x_2)^T$.

Figure 2a is an ablation study of AI-Kolmogorov with and without augmentation by clustering. The left two panels show the prediction and residuals of the most accurate expression recovered by AI-Kolmogorov ; the right two panels

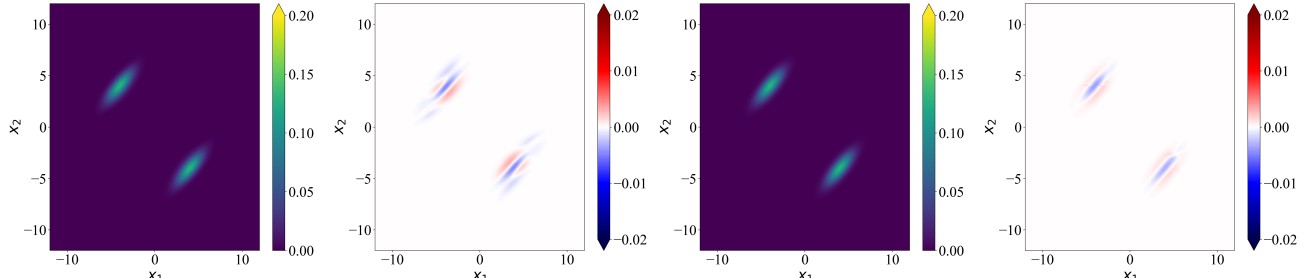

*(a)* Additive decompositon: Results on the Gaussian mixture dataset. Prediction (left) and residuals (center left) by AI-Kolmogorov . Prediction (center right) and residuals (right) by clustering augmented AI-Kolmogorov.

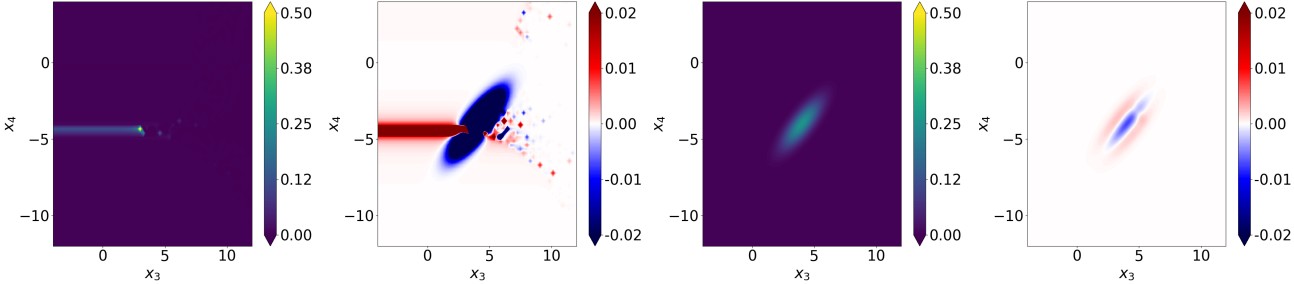

*(b)* Multiplicative decomposition: Results on the the 4-dimensional Gaussian dataset. Prediction (left) and residuals (center left) by AI-Kolmogorov. Prediction (center right) and residuals (right) by structure learning augmented AI-Kolmogorov. Only the marginal $\mathcal{N}(\mathbf{x_2} \mid \boldsymbol{\mu_2}, \boldsymbol{\Sigma})$ is shown.

*Figure 2.* Ablation study on clustering and structure learning. Top: clustering results. Bottom: structure learning results.

show the prediction and residuals of clustering augmented AI-Kolmogorov . The maximimum predicted density and absolute residual are 0.129 and 0.005 respectively (left two panels in Figure 2a), and 0.129 and 0.004 respectively (right two panels Figure 2a). The residuals of both approaches are similar in magnitude (Table 1), but the factored models yield more interpretable symbolic forms since the addition of the two component distributions is explicitly captured and the true symbolic expressions per component are simpler. The pareto fronts and recovered expressions for the Gaussian mixture dataset can be found in Appendix C.1.

We next demonstrate structure learning using independent random vectors $\mathbf{X_1}, \mathbf{X_2} \in \mathbb{R}^2$, concatenated as $\mathbf{X} \in \mathbb{R}^4$. The joint distribution is

$$f_{\mathbf{X}}(\mathbf{x}) = \mathcal{N}(\mathbf{x_1} \mid \boldsymbol{\mu_1}, \boldsymbol{\Sigma}) \cdot \mathcal{N}(\mathbf{x_2} \mid \boldsymbol{\mu_2}, \boldsymbol{\Sigma}),$$

which defines a 4D distribution whose PGM has two connected components corresponding to $\mathcal{N}(\mathbf{x_1} \mid \boldsymbol{\mu_1}, \boldsymbol{\Sigma})$ and $\mathcal{N}(\mathbf{x_2} \mid \boldsymbol{\mu_2}, \boldsymbol{\Sigma})$. We denote $\mathbf{x_1} = (x_1, x_2)^T$ and $\mathbf{x_2} = (x_3, x_4)^T$. The parameters for each component were set identically to those in the previous example.

Figure 2b is an ablation of AI-Kolmogorov against AI-Kolmogorov augmented with structure learning. To visualize the result, we only show the results for marginal $\mathcal{N}(\mathbf{x_2} \mid \boldsymbol{\mu_2}, \boldsymbol{\Sigma})$. For the panels corresponding to direct application of the pipeline to the 4D Gaussian distribution, the 2D marginal was computed by numerically integrating out

the remaining variables. Decomposition yields significantly lower error (see Table 1) for the most complex models on the pareto front. The maximimum predicted density and absolute residual are 0.630 and 0.506 respectively (left two panels of Figure 2b), and 0.258 and 0.002 respectively (right two panels of Figure 2b). For any given error level the symbolic expressions for each component are more compact and easier to interpret. While the distributions are not recovered exactly, the recovered expressions are still suggestive as they involve exponentials with negative sum of squares in the exponent. The pareto fronts and recovered expressions for the 4D Gaussian dataset can be found in Appendix C.2.

*Table 1.* MSE comparison between direct application of AI-Kolmogorov and augmented AI-Kolmogorov. The MSE with decomposition reflects the MSE after recombining the most accurate expressions from the pareto fronts

| Dataset | MSE without decomposition | MSE with decomposition |
|---|---|---|
| Gaussian mixture | $4.71 \times 10^{-4}$ | $\mathbf{3.57 \times 10^{-4}}$ |
| 4D Gaussian | $1.01 \times 10^{-3}$ | $\mathbf{4.24 \times 10^{-5}}$ |

### 4.2. Rastrigin Synthetic Dataset

We use a Rastrigin function as a synthetic exotic probability density function to demonstrate the strengths of the pipeline. The Rastrigin function is a non-convex function commonly used to benchmark optimization algorithms, but it is not

typically encountered as a probability distribution.

To adapt it into a valid distribution, we constrain its support, normalize it, and apply a bias to ensure non-negativity. Samples are drawn from this distribution using rejection sampling over the range $x_1, x_2 \in [-2, 2]$,

$$f(x) = \frac{1}{586.67}\left[10 + \sum_{i=1}^{2} 10x_i^2 - 5\cos(3\pi x_i - 6.1)\right]$$

Figure 3 illustrates the AI-Kolmogorov prediction and the residual error between the most accurate expression recovered and the ground truth. The residual plot shows that the symbolic expression recovered by AI-Kolmogorov closely matches the true density surface.

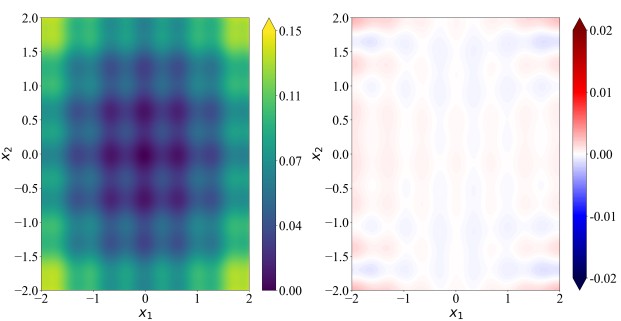

*Figure 3.* Prediction (left) and residuals with respect to the ground truth (right) of the lowest loss expression. The maximum predicted density and absolute residual are 0.140 and 0.003 respectively.

Expressions on the simpler end of the pareto front such as $1.5 + x_1^2 + x_2^2$ which captures the "coarse" bowl shape, and we recover what resembles the true expression on the more complex end of the pareto front (see Appendix C.3)

$$1.1(x_1^2 + x_2^2) - 0.56(\cos(9.4 - 12) - 0\cos(9.4x_2 + 13)) + 1.1$$

### 4.3. Muon Decay

The next example is motivated by applications in particle physics. The dataset describes muon decay, where a muon decays into an electron, and electron-antineutrino and a muon-neutrino. The features $m_{13}^2$ and $m_{23}^2$ are combinations of the momenta and energies of the outgoing particles. More specifically, the invariant mass of two-particle system where 1,2,3 labels the electron-antineutrino, the electron, and the muon-neutrino respectively. The ground-truth density (*i.e.* the differential decay width with respect to $m_{13}, m_{23}$) takes the form

$$\mathcal{N} \times \frac{(m_{23}^2 - m_\mu^2)(m_{23}^2 - m_e^2)}{(m_{13}^2 + m_{23}^2 - m_e^2 - m_\mu^2 + m_W^2)^2}$$

a rational function with a quadratic numerator in $m_{23}^2$ and a quadratic denominator in both $m_{13}^2$ and $m_{23}^2$ with normalization constant $\mathcal{N}$. We simulate samples of these events with the Monte-Carlo event generator Madgraph (Alwall et al., 2014); however, note that we have performed the simulation such that the masses of the electron ($m_e$) and muon ($m_\mu$) are comparable to the mass of the $W$-boson ($m_W$) so that there is a non-trivial dependence on $m_{13}$ in the density (in nature, $m_W \gg m_e, m_\mu$). Otherwise, the problem would be effectively 1-dimensional. This distribution could arise if there were an exotic $W'$-boson with a comparable mass to the electron and muon. The masses used for $m_\mu$, $m_e$, and $m_W$ are 78 MeV, 70 MeV, and 80.4 MeV respectively.

We apply min–max scaling to normalize the range of both features. This preserves the functional form of the distribution up to an affine transformation on each feature. Figure 4 shows the prediction and the residual error between the recovered symbolic density and the KDE. A convex hull trimming procedure (Appendix B.1) was used to estimate the support. No samples lie outside this region as it would violate energy/momentum conservation, but the support is estimated here without this domain knowledge.

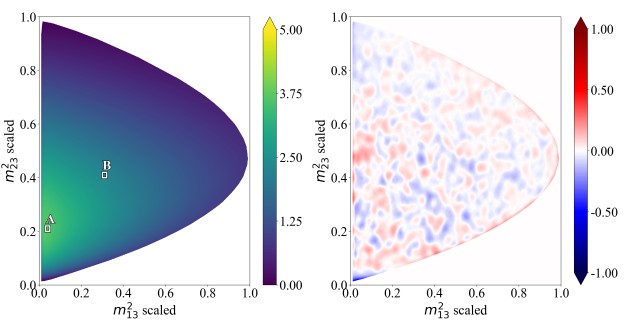

*Figure 4.* Prediction and regions for local probability mass validation indicated as A & B (left). Residuals with respect to KDE (right) of the lowest loss expression. The maximum predicted density is 3.677, and the maximum absolute residual is 0.640.

**Muon decay local probability mass validation**  To avoid relying on the availability of a ground truth expression to validate the KDE and symbolic density estimate (the most accurate expression recovered) we instead examine the probability mass over two small square regions (Figure 4). The empirical probability mass over each square region is obtained by finding the fraction of raw samples that are present inside each square. We then integrate both the KDE and symbolic density estimate numerically over these regions to compute an estimated probability mass. Finally, we compare the estimated probability mass from the density estimates to the empirical probability mass to assess if they fit the true distribution well.

The results in Table 2 suggest that the density estimates are accurate in the two regions examined. The pareto front of re-

*Table 2.* Muon decay local probability mass validation on two $0.02 \times 0.02$ square regions.

| Method | Region A | Region B |
|---|---|---|
| Empirical | $1.4960 \times 10^{-3}$ | $9.3200 \times 10^{-4}$ |
| KDE | $1.4811 \times 10^{-3}$ | $9.1705 \times 10^{-4}$ |
| SR | $1.4638 \times 10^{-3}$ | $9.1224 \times 10^{-4}$ |

covered expressions is reported in Appendix C.4. Although the exact form is not recovered, the resulting expressions achieve very low error with respect to the KDE within the support of the distribution.

### 4.4. Dijet

To test our framework on a more challenging dataset, we simulate the production of two jets in proton-proton collisions at the Large Hadron Collider. This process depends in part on the content of the proton, the "parton distribution function" (we use NNPDF2.3 (Ball et al., 2013)) as well as the model of hadronization. The parton distribution cannot be calculated from first principles and must instead be derived from experiment with an assumed model. A variety of models are assumed (Dulat et al., 2016; Ball et al., 2015), and thus can represent a bias that must be considered (Gao & Nadolsky, 2014); a similar statement could be said of the hadronization model, but we perform our analysis at parton-level. One could thus imagine using our symbolic regression to either gain insight into the underlying truth distribution, or point towards more promising and less-considered functional forms. There are many observables that one could use to gain insight, we choose two high-level observables which are calculated from the simulation: the invariant mass of the jet pair, $m_{jj}$, and the scalar sum of the transverse momenta, $H_T = \sum_{1,2} |p_T^i|$. Note that these observables are chosen somewhat arbitrarily and are not optimized for the problem at hand. Our goal is to test our framework on a problem where the understanding of underlying distribution is on less solid footing.

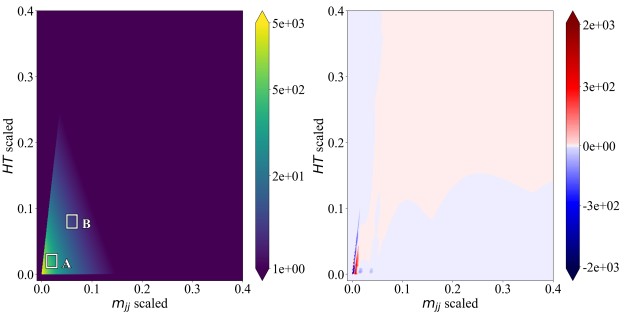

*Figure 5.* Prediction and regions for local probability mass validation indicated as A & B (left). Residuals with respect to NSF (right) of the lowest loss expression. The maximum predicted density is 4555.25 and the maximum absolute residual is 1789.14.

This dataset set is particularly challenging because of the presence of sharp discontinuities at the boundary of the support and because the distribution is concentrated in a small region. Figure 5 shows the prediction and the residual error between the recovered symbolic density and the NSF density estimate. The residuals reported are large, though this may be attributed to the sharp discontinuity near the distribution peak. The heat map of the prediction is qualitatively similar to the distribution of raw samples.

**Dijet local probability mass validation**  Similarly to the previous example of Muon Decay, we examine the probability mass over two small square regions (Figure 5). The results in Table 3 suggest that the density estimate obtained by NSF outperforms KDE. This can also be seen in the log-likelihood scores in Figure 6 .The SR stage uses the NSF density estimate, as the accuracy of KDE would otherwise have been a bottleneck to the method.

*Table 3.* Dijet local probability mass validation on two $0.02 \times 0.02$ square regions.

| Method | Region A | Region B |
|---|---|---|
| Empirical | $1.0270 \times 10^{-1}$ | $2.3200 \times 10^{-3}$ |
| KDE | $6.7234 \times 10^{-2}$ | $1.5040 \times 10^{-2}$ |
| NSF | $1.0974 \times 10^{-1}$ | $1.9397 \times 10^{-3}$ |
| SR | $1.1493 \times 10^{-1}$ | $2.2609 \times 10^{-3}$ |

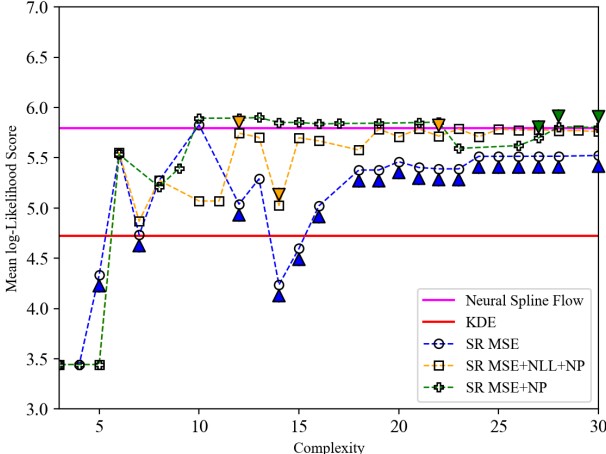

*Figure 6.* Mean log-likelihood scores against expression complexity for SR using only MSE loss and SR using MSE, mean negative log-likelihood (NLL), and penalties for negative density predictions (NP) in the loss function. Triangles indicate expressions that made negative predictions on test samples.

Figure 6 shows the mean log-likelihood scores of the expressions discovered by AI-Kolmogorov using only MSE loss versus a combination of MSE loss, mean negative log-likelihood, and penalties for negative predictions. The volume under each expression was normalized prior to com-

puting their mean log-likelihood scores. The results suggest that loss function design may improve both accuracy and validity of the returned expressions. Accurate models may be discovered using MSE loss only, though these are more likely to return negative predictions over the test samples (indicated by triangles on the markers in Figure 6).

Although the log-likelihood scores appear to drop off for expressions with complexity greater than 10 in Figure 6 for SR MSE this was due to some of them producing small negative predictions on a small subset of the test set (at most 282 out of 10,000 test samples). The density predictions are clipped if they fall below a tiny threshold, and their log-likelihood scores are sensitive to the selected clipping threshold. Tuning the constants of the recovered expressions with an additional bias term could be done to satisfy the non-negativity constraint and potentially improve the log-likelihood score. Augmenting the loss function with the negative log-likelihood (SR MSE + NLL + NP) does not appear to have a significant impact on the performance of SR nor does it perform as well as SR with only the MSE and negative prediction penalties (SR MSE + NP) with respect to the log-likelihood metric on this particular problem. Nevertheless, it injects context of the original density estimation problem into AI-Kolmogorov's SR stage, and may still be useful on other problems with appropriately tuned weights. SR MSE + NP appears to have a slight improvement over the NSF, however, these expressions do not strictly satisfy the constraints of a valid distribution, while NSF does by construction. These expressions may nevertheless have terms useful for interpretability.

Finally, an attempt was made using SR without MSE and only the negative log-likelihood loss and the negative prediction penalty. This loss function produced expressions that did not fit the distribution well, achieving a maximum mean log-likelihood score of -0.12, consequently it does not appear on the scale in Figure 6

The pareto front of recovered expressions is reported in Appendix C.5. The loss function only softly enforces normalization via MSE loss against the NSF density estimate over a regular grid in the support.

### 4.5. Key Insights and Observations

**Clustering (additive decomposition):** Identifies well-separated modes and fits unimodal components before re-combination by summation; interpretability improves without sacrificing accuracy.

**Structure learning (multiplicative decomposition):** Exploits independence between subsets of the covariates to reduce dimensionality at the SR stage; this yields markedly lower errors and more compact expressions than non-decomposed baselines.

**Support Estimation:** Improves stability and interpretability; meaningful symbolic descriptions may only be valid over the support without sharp discontinuities in the target probability density function.

**Nonparametric density estimate:** Poor nonparametric density estimates bottleneck the performance of AI-Kolmogorov's SR stage.

**Loss function design:** MSE against the nonparametric density estimate over a regular grid is typically sufficient to obtain accurate models although additional loss function design has benefits. Raw samples and log-likelihood alone may result in invalid models as expressions that predict negative densities or large density in regions where samples are sparse are not penalized. With GP-based SR, the loss function need not be differentiable, however, other methods may require this property.

**Exploration of symbolic expressions:** With limited domain specific context, a diversity of candidate models may inform scientists of interesting structure present in the raw samples.

## 5. Future Work and Conclusion

There are several possible extensions to AI-Kolmogorov for SymDE that warrant further investigation. The current approach provides a solid foundation by demonstrating the feasibility of combining decomposition strategies with state-of-the-art symbolic regression techniques. Future work could explore more sophisticated decomposition strategies. The incorporation of the likelihood, computed on the raw input samples, directly into the loss has shown promise. With sufficient complexity, it outperforms the neural density estimator. However, labels from the neural density estimator helps stabilize the search algorithm. Future work could incorporate the normalization requirement of densities though a Monte-Carlo integration into the loss. Agentic AI frameworks for SymDE are also a promising avenue for future work.

Equally important is the development of principled diagnostics to assess goodness-of-fit, especially for heavy-tailed distributions and small-sample settings. Real-world distributions may have heavy tails and understanding the performance and understanding the limitations of our tool in small sample regime would be necessary. Applying AI-Kolmogorov to real-world problems in physics, finance, and other domains would provide valuable insights with practical utility, and help the development of domain-specific and interpretable statistical models.

## Impact Statement

This paper presents a work whose goal is to advance the field of Machine Learning. There are many potential societal consequences of our work, none which we feel must be specifically highlighted here.

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

# A. Appendix: Mathematical Foundations

### A.1. Probabilistic Graphical Models

Any probability distribution can be represented as a directed acyclic graph (DAG) where nodes represent random variables and links represent probabilistic relationships. The joint distribution factorizes as $p(x_1, ..., x_n) = p(x_n|x_1, ..., x_{n-1})...p(x_2|x_1)p(x_1)$.

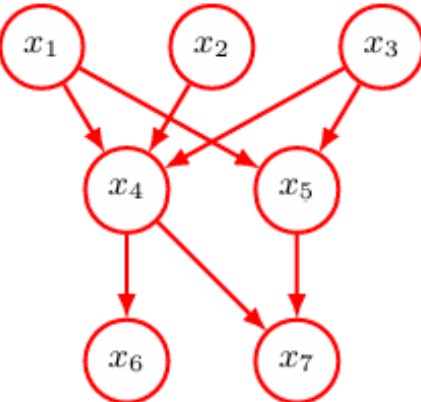

*Figure 7.* Example PGM for distribution $p(x_1, ..., x_7) = p(x_1)p(x_2)p(x_3)p(x_4|x_1, x_2, x_3)$ $\cdot p(x_5|x_1, x_3)p(x_6|x_4)p(x_7|x_4, x_5)$ (Bishop & Bishop, 2023).

# B. Appendix: Algorithmic Details

### B.1. Support Estimation

The support of a probability distribution is $\text{supp}(f_\mathbf{X}) = \{\mathbf{x} \in \mathbb{R}^d | f_\mathbf{X}(\mathbf{x}) > 0\}$. Convex hull estimates improve boundary handling for bounded distributions.

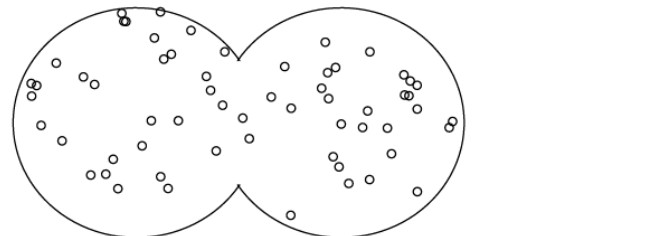 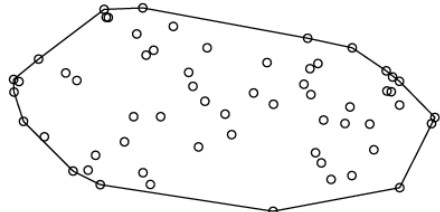

*Figure 8.* Convex hull support estimates. Left: true support. Right: convex hull estimate (Cui, 2021).

To correct bias of non-parametric density estimates at known boundaries of a distribution we can employ the reflection trick as illustrated in Figure 9.

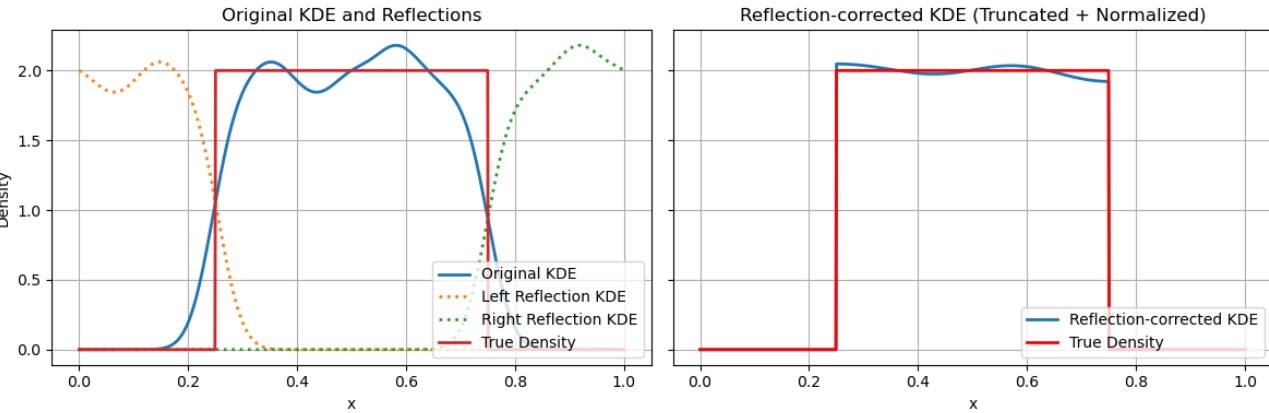

Figure 9. Reflection trick in one dimension.

## B.2. Clustering and Structure Learning

DBSCAN clustering automatically infers cluster count and handles non-parametric cluster shapes. Structure learning discovers independent variable subsets, enabling problem decomposition via the product rule.

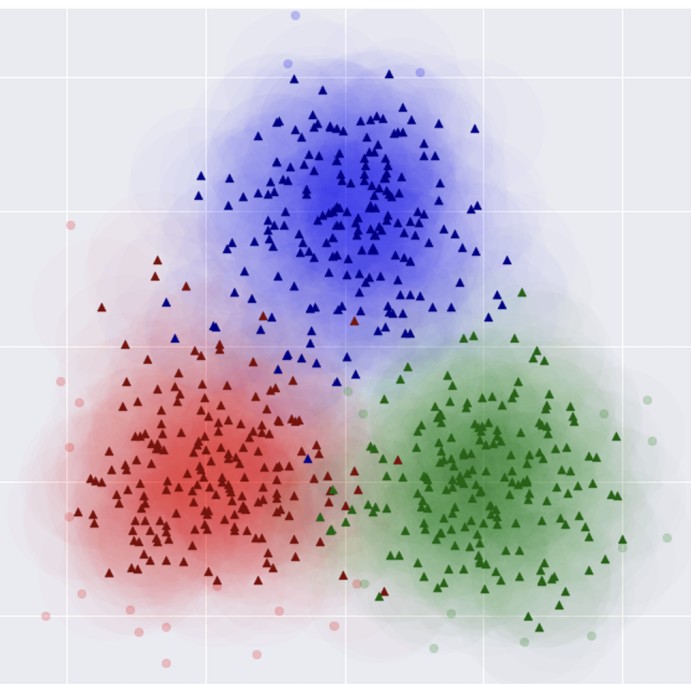

Figure 10. DBSCAN on a 2D Gaussian mixture. Core points have $\epsilon$-neighborhoods with at least minPts samples (Jang & Jiang, 2019).

## B.3. Kernel Density Estimation

KDE estimates density as $\hat{f}_n(x) = \frac{1}{nh^d} \sum_{i=1}^{n} K\left(\frac{x-X_i}{h}\right)$ where $h$ is bandwidth and $K$ is kernel function. The reflection trick improves boundary estimates by reflecting samples across boundaries.

### B.3.1. FAST FOURIER TRANSFORM KDE

Fast Fourier Transform (FFT) KDE is a computationally efficient approach that leverages the convolution theorem to accelerate density estimation. Instead of directly computing the sum of kernel functions at each evaluation point, FFT KDE

discretizes the domain into a regular grid and performs the convolution in the frequency domain. This approach transforms the computational complexity from $O(nm)$ to $O(n \log n + m \log m)$, where $n$ is the number of samples and $m$ is the number of grid points, making it particularly advantageous for large datasets and fine-grained density estimates. The method works by: (1) creating a histogram of the data on a regular grid, (2) computing the FFT of both the histogram and the kernel, (3) multiplying them in the frequency domain, and (4) applying the inverse FFT to obtain the final density estimate. This technique is especially effective for multivariate data and is widely used in modern statistical software packages.

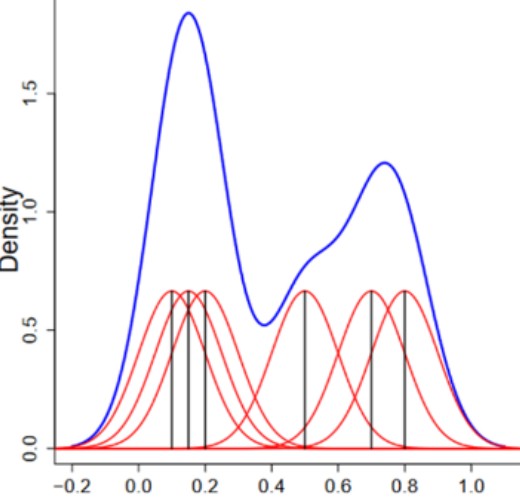

*Figure 11.* Kernel Density Estimation in 1D (Chen, 2017)

## C. Appendix: Experimental Results

### C.1. Gaussian Mixture Dataset

*Table 4.* Hyper-parameters for symbolic regression on the Gaussian mixture dataset

| Hyper-parameter | Value | Notes |
|---|---|---|
| **Kernel Density Estimation** | | |
| number of samples | 450000 | |
| bandwidth | 0.101 | KDE bandwidth selected using cross-validation. |
| **PySR** | | |
| binary operators | [+, -, *, /] | |
| unary operators | [exp, log, pow2, pow3] | Allowed unary operators. |
| maxsize | 50 | Maximum size of an expression tree. |
| ncycles per iteration | 380 | Cycles per iteration of the genetic algorithm. |
| parsimony | 0.001 | Penalty for complexity of expressions. |
| adaptive parsimony scaling | 1040 | Scales the penalty based on frequency of each complexity level |
| niterations for SR | 8000 | Iterations for SR. |
| num populations for SR | 15 | Number of populations for SR. |
| population size for SR | 30 | Number of expressions per population in SR |
| elementwise loss | MSE | Loss function used for SR. |
| batch size | 128 | batch size of the dataset used to evaluate loss of expressions |

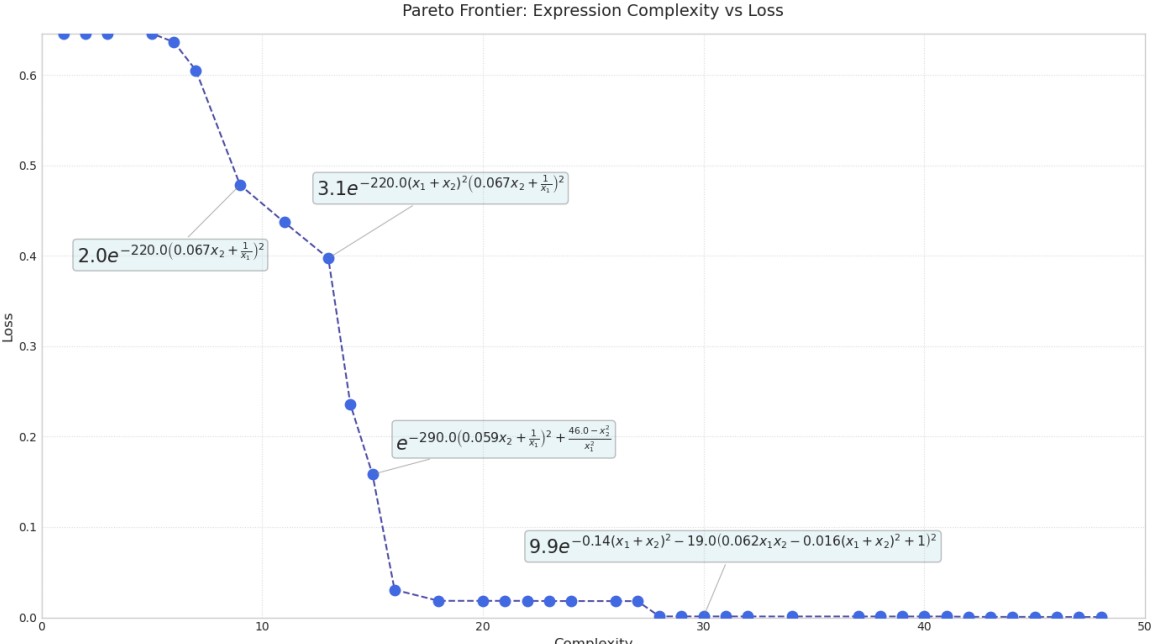

*Figure 12.* Pareto front of recovered expressions for the Gaussian mixture dataset.

*Table 5.* Expressions simplified with *sympy* alongside their raw complexity score from PySR for the Gaussian mixture dataset.

| Raw Complexity | Simplified Expression |
|---|---|
| 1 | $0.13$ |
| 2 | $0.14$ |
| 3 | $0.13$ |
| 5 | $0.14$ |
| 6 | $0.24 - 0.0023x_2^2$ |
| 7 | $e^{-0.23(x_1+x_2)^2}$ |
| 9 | $2e^{-220\left(0.067x_2+\frac{1}{x_1}\right)^2}$ |
| 11 | $2.5e^{-16(0.062x_1x_2+1)^2}$ |
| 13 | $3.1e^{-\frac{220(x_1+x_2)^2(0.067x_1x_2+1)^2}{x_1^2}}$ |
| 14 | $13e^{-(x_1+x_2)^2-260\left(0.062x_2+\frac{1}{x_1}\right)^2}$ |
| 15 | $e^{\frac{-x_2^2-290(0.059x_1x_2+1)^2+46}{x_1^2}}$ |
| 16 | $9.5e^{-0.15(x_1+x_2)^2-220\left(0.067x_2+\frac{1}{x_1}\right)^2}$ |
| 18 | $9.9e^{-0.15(x_1+x_2)^2-18(0.064x_1x_2+1)^2}$ |
| 20 | $9.9e^{-0.15(x_1+x_2)^2-18(0.064x_1x_2+1)^2}+0.0052$ |
| 21 | $9.9e^{-0.15(x_1+x_2)^2-19(0.062x_1x_2+1)^2}+0.0038$ |
| 22 | $9.9e^{-0.15(x_1+x_2)^2-19(0.062x_1x_2+1)^2}+0.0041$ |
| 23 | $9.9e^{-18(0.064x_1x_2+1)^2-0.29\left(0.71x_1+0.71x_2+\frac{1}{x_1^3}\right)^2}$ |
| 24 | $9.9e^{-19(0.062x_1x_2+1)^2-0.45\left(0.58x_1+0.58x_2+\frac{1}{x_1^3}\right)^2}$ |
| 26 | $9.8e^{-19(0.062x_1x_2+1)^2-0.54\left(0.53x_1+0.53x_2+\frac{1}{x_1^3}\right)^2}$ |
| 27 | $9.8e^{-18(0.065x_1x_2+1)^2-0.54\left(0.53x_1+0.53x_2+\frac{1}{x_1^3}\right)^2}$ |
| 28 | $9.9e^{-0.14(x_1+x_2)^2-19\left(0.062x_1x_2-0.016(x_1+x_2)^2+1\right)^2}$ |
| 29 | $9.9e^{-0.14(x_1+x_2)^2-20\left(0.062x_1x_2-0.015(x_1+x_2)^2+1\right)^2}$ |
| 30 | $9.9e^{-0.14(x_1+x_2)^2-19\left(0.062x_1x_2-0.016(x_1+x_2)^2+1\right)^2}$ |
| 31 | $9.9e^{-0.14(x_1+x_2)^2-19\left(0.062x_1x_2-0.016(x_1+x_2)^2+1\right)^2}$ |
| 32 | $9.9e^{-0.14(x_1+x_2)^2-19\left(0.062x_1x_2-0.016(x_1+x_2)^2+1\right)^2}$ |
| 34 | $10e^{-0.14(x_1+x_2)^2-19\left(0.062x_1x_2-0.016(x_1+x_2)^2+1\right)^2}$ |
| 37 | $10e^{-0.14(x_1+x_2)^2-20\left(-0.064x_1x_2+0.015(x_1+x_2)^2-1\right)^2}$ |
| 38 | $10e^{-0.14(x_1+x_2)^2-20\left(0.064x_1x_2-0.015(x_1+x_2)^2+1\right)^2}$ |
| 39 | $10e^{-0.14(x_1+x_2)^2-20\left(0.064x_1x_2-0.015(x_1+x_2)^2+1\right)^2}$ |
| 40 | $10e^{-0.14(x_1+x_2)^2-19\left(0.05x_2(1.3x_1-0.003)-0.015(x_1+x_2)^2+1\right)^2}$ |
| 41 | $10e^{-0.14(x_1+x_2)^2-19\left(-0.05x_2(1.3x_1-0.0027)+0.015(x_1+x_2)^2-1\right)^2}$ |
| 42 | $9.9e^{-0.14(x_1+x_2)^2-91\left(-0.002x_1^2+0.015x_1x_2+\frac{0.15x_1}{x_2}-0.0073x_2^2+0.48+\frac{1}{x_2^2}\right)^2}$ |
| 43 | $9.9e^{-0.14(x_1+x_2)^2-83\left(-0.0023x_1^2+0.015x_1x_2+\frac{0.15x_1}{x_2}-0.0079x_2^2+0.5+\frac{1}{x_2^2}\right)^2}$ |
| 44 | $9.9e^{-0.14(x_1+x_2)^2-83\left(0.0056x_1^2+0.031x_1x_2+\frac{0.15x_1}{x_2}-0.0078(x_1+x_2)^2+0.5+\frac{1}{x_2^2}\right)^2}$ |
| 45 | $9.9e^{-0.14(x_1+x_2)^2-83\left(-0.0023x_1^2+0.015x_1x_2+\frac{0.15x_1}{x_2}-0.0078x_2^2+0.5+\frac{1}{x_2^2}\right)^2}$ |
| 46 | $9.8e^{-0.13(x_1+x_2)^2-\frac{4.5\left(0.032x_2(0.16x_1-x_2)(x_1+x_2)^2-(0.13x_1x_2+2.1)((0.13x_1+x_2)(0.16x_1-x_2)-2.1)\right)^2}{x_2^2(0.16x_1-x_2)^2}}$ |
| 47 | $9.5e^{-0.13(x_1+x_2)^2-\frac{4.5\left(0.032x_2(0.19x_1-x_2)(x_1+x_2)^2-(0.13x_1x_2+2.1)((0.13x_1+x_2)(0.19x_1-x_2)-2.1)\right)^2}{x_2^2(0.19x_1-x_2)^2}}$ |
| 48 | $9.9e^{-0.14(x_1+x_2)^2-\frac{4.6\left(0.033x_2(0.19x_1-x_2)(x_1+x_2)^2-(0.13x_1x_2+2.1)((0.13x_1+x_2)(0.19x_1-x_2)-2.1)\right)^2}{x_2^2(0.19x_1-x_2)^2}}$ |

*Table 6.* Hyper-parameters for symbolic regression on each cluster

| Hyper-parameter | Value | Notes |
|---|---|---|
| **DBSCAN Clustering** | | |
| eps | 5 | Maximum Neighborhood radius. |
| min samples | 10 | Neighborhood minimum size. |
| **Kernel Density Estimation** | | |
| number of samples | 225000 | |
| bandwidth | 0.0923 | KDE bandwidth selected using cross-validation. |
| **PySR** | | |
| binary operators | [+, -, *, /] | |
| unary operators | [exp, log, pow2, pow3] | Allowed unary operators. |
| maxsize | 50 | Maximum size of an expression tree. |
| ncycles per iteration | 380 | Cycles per iteration of the genetic algorithm. |
| parsimony | 0.001 | Penalty for complexity of expressions. |
| adaptive parsimony scaling | 1040 | Scales the penalty based on frequency of each complexity level |
| niterations for SR | 8000 | Iterations for SR. |
| num populations for SR | 15 | Number of populations for SR. |
| population size for SR | 30 | Number of expressions per population in SR |
| elementwise loss | MSE | Loss function used for SR. |
| batch size | 128 | batch size of the dataset used to evaluate loss of expressions |

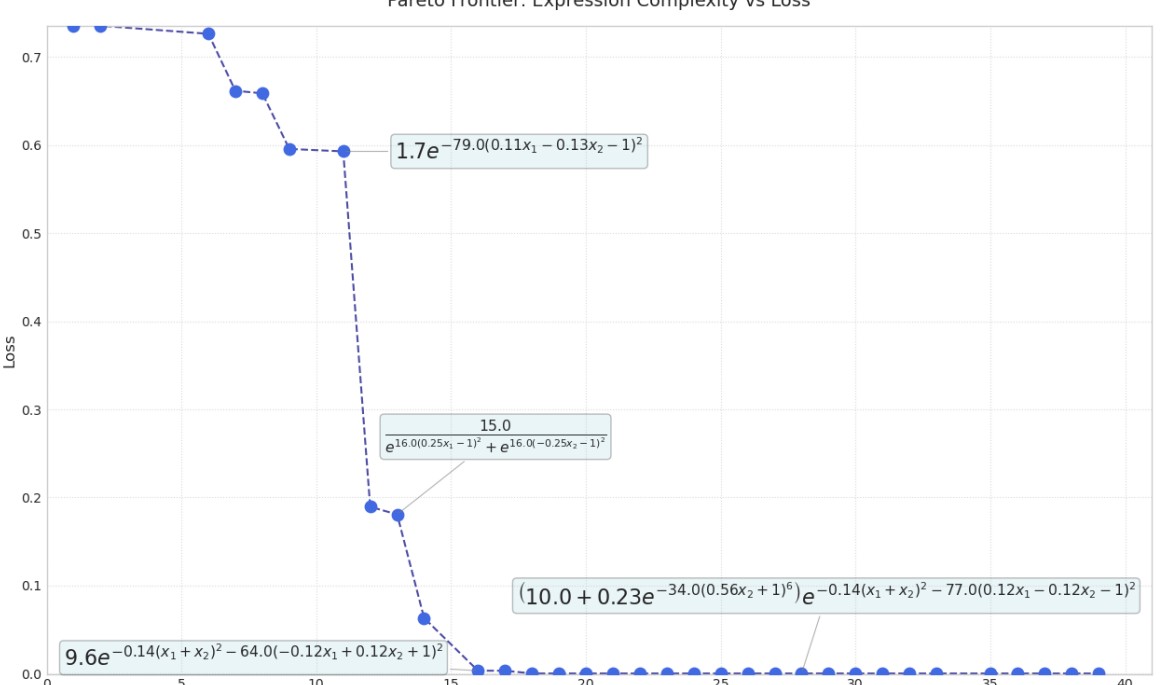

*Figure 13.* Pareto front of recovered expressions for the Gaussian cluster 1 dataset.

*Table 7.* Expressions simplified with *sympy* alongside their raw complexity score for the Gaussian cluster 1 dataset.

| Raw Complexity | Simplified Expression |
| --- | --- |
| 1 | $0.15$ |
| 2 | $0.15$ |
| 6 | $\dfrac{x_1}{x_1^2 + 6.6}$ |
| 7 | $1.1e^{-16(-0.25x_2-1)^2}$ |
| 8 | $\dfrac{1}{16(0.25x_2+1)^2 + 1}$ |
| 9 | $1.5e^{-64(0.12x_1-0.12x_2-1)^2}$ |
| 11 | $1.7e^{-79(0.11x_1-0.13x_2-1)^2}$ |
| 12 | $9.6e^{-16(0.25x_1-1)^2-16(0.25x_2+1)^2}$ |
| 13 | $\dfrac{15}{e^{16(0.25x_1-1)^2} + e^{16(-0.25x_2-1)^2}}$ |
| 14 | $11e^{-16(-0.25x_2-1)^2-64(0.12x_1-0.12x_2-1)^2}$ |
| 16 | $9.6e^{-0.14(x_1+x_2)^2-64(-0.12x_1+0.12x_2+1)^2}$ |
| 17 | $9.6e^{-0.14(x_1+x_2)^2-64(0.12x_1-0.12x_2-1)^2}$ |
| 18 | $10e^{-0.14(x_1+x_2)^2-77(0.12x_1-0.12x_2-1)^2}$ |
| 19 | $10e^{-0.14(x_1+x_2)^2-77(0.12x_1-0.12x_2-1)^2}$ |
| 20 | $10e^{-0.14(x_1+x_2)^2-77(0.12x_1-0.12x_2-1)^2}$ |
| 21 | $10e^{-77(0.12x_1-0.12x_2-1)^2-0.14(x_1+x_2-0.0015)^2}$ |
| 22 | $10e^{-0.14(1x_1+x_2)^2-77(0.12x_1-0.12x_2-1)^2}$ |
| 23 | $\left(e^{10.4x_2} + 10\right)e^{-0.14(x_1+x_2)^2-77(0.12x_1-0.12x_2-1)^2}$ |
| 24 | $\dfrac{10e^{00.58x_2^2}}{e^{00.58x_2^2+0.14(x_1+x_2)^2+77(0.12x_1-0.12x_2-1)^2} - 1}$ |
| 25 | $\dfrac{10e^{00.55x_2^2}}{e^{00.55x_2^2+0.14(x_1+x_2)^2+77(0.12x_1-0.12x_2-1)^2} - 1}$ |
| 26 | $\dfrac{10e^{00.58x_2^2}}{e^{00.58x_2^2+0.14(x_1+x_2)^2+77(0.12x_1-0.12x_2-1)^2} - 1.3}$ |
| 27 | $\left(10e^{34(0.56x_2+1)^6} + 0.16\right)e^{-0.14(x_1+x_2)^2-34(0.56x_2+1)^6-77(0.12x_1-0.12x_2-1)^2}$ |
| 28 | $\left(10e^{34(0.56x_2+1)^6} + 0.23\right)e^{-0.14(x_1+x_2)^2-34(0.56x_2+1)^6-77(0.12x_1-0.12x_2-1)^2}$ |
| 29 | $\left(10e^{34(-0.56x_2-1)^6} + 0.23\right)e^{-0.14(x_1+x_2)^2-34(-0.56x_2-1)^6-77(0.12x_1-0.12x_2-1)^2}$ |
| 30 | $\left(10e^{10.96(0.71x_2(x_2+2)-1)^2} + 0.26\right)e^{-0.14(x_1+x_2)^2-2(0.71x_2(x_2+2)-1)^2-77(0.12x_1-0.12x_2-1)^2}$ |
| 31 | $\left(10e^{10.96(0.71x_2(x_2+2)-1)^2} + 0.26\right)e^{-0.14(x_1+x_2)^2-2(0.71x_2(x_2+2)-1)^2-77(0.12x_1-0.12x_2-1)^2}$ |
| 32 | $\left(10e^{20.25(0.67x_2(x_2+2)-1)^2} + 0.26\right)e^{-0.14(x_1+x_2)^2-2.2(0.67x_2(x_2+2)-1)^2-77(0.12x_1-0.12x_2-1)^2}$ |
| 33 | $\left(10e^{20.56(0.62x_2(x_2+2)-1)^2} + 0.26\right)e^{-0.14(1x_1+x_2)^2-2.6(0.62x_2(x_2+2)-1)^2-77(0.12x_1-0.12x_2-1)^2}$ |
| 35 | $\left(10e^{20.25(0.67x_2(x_2+2)-1)^2} + 0.26\right)e^{-0.14(1x_1+x_2)^2-2.2(0.67x_2(x_2+2)-1)^2-77(0.12x_1-0.12x_2-1)^2}$ |
| 36 | $\left(10e^{20.56(0.62x_2(x_2+1.7)-1)^2} + 0.27\right)e^{-0.14(1x_1+x_2)^2-2.6(0.62x_2(x_2+1.7)-1)^2-77(0.12x_1-0.12x_2-1)^2}$ |
| 37 | $\left(10e^{20.56(0.62x_2(x_2+1.7)-1)^2} + 0.29\right)e^{-0.14(1x_1+x_2)^2-2.6(0.62x_2(x_2+1.7)-1)^2-77(0.12x_1-0.12x_2-1)^2}$ |
| 38 | $\left(10e^{20.56(0.62x_2(x_2+1.7)-1)^2} + 0.27\right)e^{-0.14(1x_1+x_2)^2-2.6(0.62x_2(x_2+1.7)-1)^2-77(0.12x_1-0.12x_2-1)^2}$ |
| 39 | $\left(10e^{20.56(0.62x_2(x_2+1.7)-1)^2} + 0.29\right)e^{-0.14(1x_1+x_2)^2-2.6(0.62x_2(x_2+1.7)-1)^2-77(0.12x_1-0.12x_2-1)^2}$ |

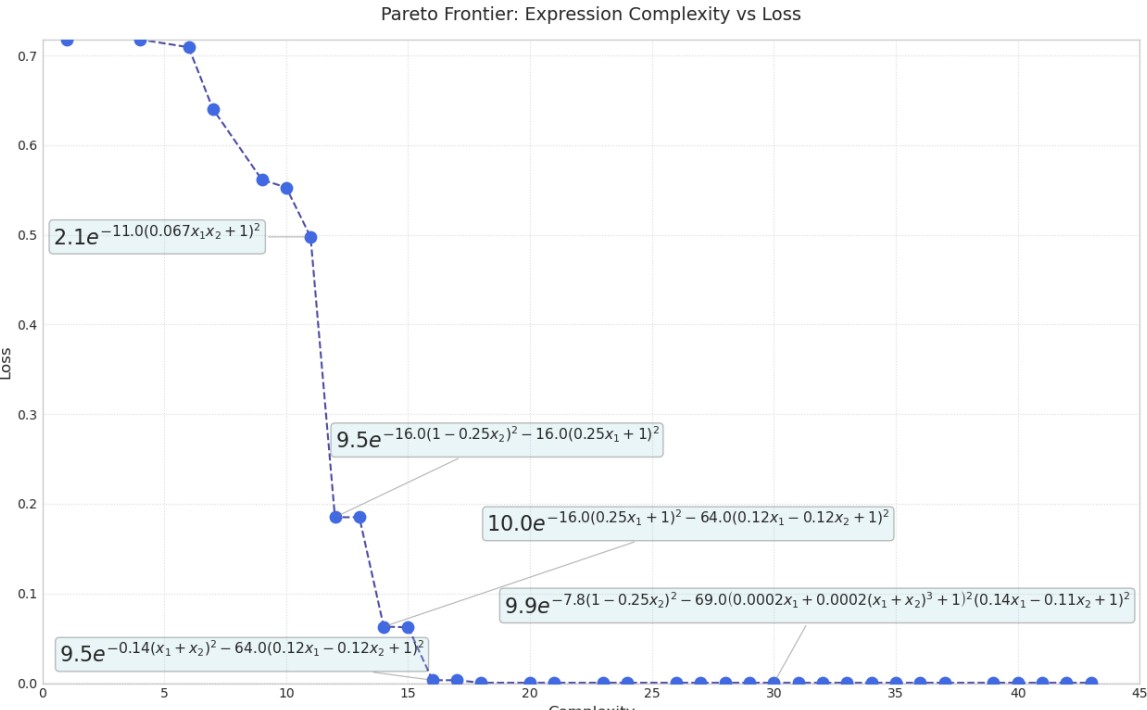

*Figure 14.* Pareto front of recovered expressions for the Gaussian cluster 2 dataset.

*Table 8.* Expressions simplified with *sympy* for the Gaussian Cluster 2 dataset.

| Raw Complexity | Simplified Expression |
|---|---|
| 1 | $0.15$ |
| 4 | $0.15$ |
| 6 | $\dfrac{0.13}{e^{x_1} + 0.61}$ |
| 7 | $e^{-8.5(0.25x_1+1)^2}$ |
| 9 | $e^{-6.9(0.062x_1x_2+1)^2}$ |
| 10 | $e^{-660(0.062x_1x_2+1)^6}$ |
| 11 | $2.1\,e^{-11(0.067x_1x_2+1)^2}$ |
| 12 | $9.5\,e^{-16(1-0.25x_2)^2-16(0.25x_1+1)^2}$ |
| 13 | $9.5\,e^{-16(1-0.25x_2)^2-16(0.25x_1+1)^2}$ |
| 14 | $10\,e^{-16(0.25x_1+1)^2-64(0.12x_1-0.12x_2+1)^2}$ |
| 15 | $9.5\,e^{-16(0.25x_1+1)^2-64(0.12x_1-0.12x_2+1)^2}$ |
| 16 | $9.5\,e^{-0.14(x_1+x_2)^2-64(0.12x_1-0.12x_2+1)^2}$ |
| 17 | $9\,e^{-0.14(x_1+x_2)^2-64(0.12x_1-0.12x_2+1)^2}$ |
| 18 | $9.9\,e^{-0.14(x_1+x_2)^2-76(0.13x_1-0.13x_2+1)^2}$ |
| 20 | $8.1\,e^{-0.14(x_1+x_2)^2-77(0.12x_1-0.12x_2+1)^2}$ |
| 21 | $8\,e^{-0.14(x_1+x_2)^2-77(0.12x_1-0.12x_2+1)^2}$ |
| 23 | $8\,e^{-0.14(x_1+x_2)^2-77(0.12x_1-0.12x_2+1)^2}$ |
| 24 | $9.9\,e^{-7.8(0.25x_2-1)^2-69(-0.14x_1+0.11x_2-1)^2}$ |
| 26 | $9.9\,e^{-7.8(0.25x_2-1)^2-\dfrac{0.062\left((x_2-4)\left(-1.2x_1+0.92x_2-8.3\right)-1.8e-04\right)^2}{(0.25x_2-1)^2}}$ |
| 27 | $9.9\,e^{-7.8(0.25x_2-1)^2-69\left(2.0e-04(x_1+x_2)^3+1\right)^2(0.14x_1-0.11x_2+1)^2}$ |
| 28 | $9.9\,e^{-7.8(0.25x_2-1)^2-69\left(2.0e-04(x_1+x_2)^3+1\right)^2(0.14x_1-0.11x_2+1)^2}$ |
| 29 | $9.9\,e^{-7.8(0.25x_2-1)^2-69\left(2.0e-04(x_1+x_2+0.39)^3+1\right)^2(0.14x_1-0.11x_2+1)^2}$ |
| 30 | $9.9\,e^{-7.8(1-0.25x_2)^2-69\left(2.0e-04x_1+2.0e-04(x_1+x_2)^3+1\right)^2(0.14x_1-0.11x_2+1)^2}$ |
| 31 | $9.9\,e^{-7.8(0.25x_2-1)^2-67\left(2.4e-04(0.79x_1+x_2)^3+1\right)^2(0.14x_1-0.11x_2+1)^2}$ |
| 32 | $9.9\,e^{-7.8(0.25x_2-1)^2-68\left(2.0e-04(x_1+x_2+0.79)^3+1\right)^2(0.14x_1-0.11x_2+1)^2}$ |
| 33 | $9.9\,e^{-67\left(1+\dfrac{0.0029\left(0.43x_2\left(x_1+x_2\right)+1\right)^3}{x_2^3}\right)^2(0.14x_1-0.11x_2+1)^2-7.8(0.25x_2-1)^2}$ |
| 34 | $9.9\,e^{-68\left(1+\dfrac{0.015\left(0.25x_2\left(x_1+x_2\right)+1\right)^3}{x_2^3}\right)^2(0.14x_1-0.11x_2+1)^2-7.8(0.25x_2-1)^2}$ |
| 35 | $9.9\,e^{-7.8(1-0.25x_2)^2-67(0.14x_1-0.11x_2+1)^2\left(-2.4e-04x_2+1+\dfrac{0.015\left(0.25x_2\left(x_1+x_2\right)+1\right)^3}{x_2^3}\right)^2}$ |
| 36 | $9.9\,e^{-7.8(0.25x_2-1)^2-68(0.14x_1-0.11x_2+1)^2\left(2.4e-04x_2-1-\dfrac{0.015\left(0.25x_2\left(x_1+x_2\right)+1\right)^3}{x_2^3}\right)^2}$ |
| 37 | $9.9\,e^{-7.8(0.25x_2-1)^2-69(0.14x_1-0.11x_2+1)^2\left(-7.2e-04x_2+1+\dfrac{0.015\left(0.25x_2\left(x_1+x_2\right)+1\right)^3}{x_2^3}\right)^2}$ |
| 39 | $9.9\,e^{-7.8(0.25x_2-1)^2-69(0.14x_1-0.11x_2+1)^2\left(-7.4e-04x_2+1+\dfrac{0.015\left(0.25x_2\left(x_1+x_2\right)+1\right)^3}{x_2^3}\right)^2}$ |
| 40 | $\left(9.2-0.92\log\left(\left(0.61e^{00.48x_1^2}+1\right)e^{-0.48x_1^2}\right)\right)e^{-7.8(0.25x_2-1)^2-\dfrac{0.062\left((x_2-4)\left(1.2x_1-0.92x_2+8.3\right)+1.8e-04\right)^2}{(0.25x_2-1)^2}}$ |
| 41 | $\left(10-0.92e^{-0.008x_1^6}\right)e^{-7.8(0.25x_2-1)^2-\dfrac{0.062\left((x_2-4)\left(-1.2x_1+0.92x_2-8.3\right)-1.8e-04\right)^2}{(0.25x_2-1)^2}}$ |
| 42 | $\left(9.9-0.75e^{-0.008x_1^6}\right)e^{-7.8(0.25x_2-1)^2-\dfrac{0.062\left((x_2-4)\left(-1.2x_1+0.92x_2-8.3\right)-2.3e-04\right)^2}{(0.25x_2-1)^2}}$ |
| 43 | $\left(9.9-0.61e^{-0.008x_1^6}\right)e^{-7.8(0.25x_2-1)^2-\dfrac{0.062\left((x_2-4)\left(-1.2x_1+0.92x_2-8.3\right)-1.8e-04\right)^2}{(0.25x_2-1)^2}}$ |

## C.2. 4D Gaussian Dataset

*Table 9.* Hyper-parameters for symbolic regression on the 4D Gaussian dataset

| Hyper-parameter | Value | Notes |
|---|---|---|
| **Kernel Density Estimation** | | |
| number of samples | 225000 | |
| bandwidth | 0.0857 | KDE bandwidth selected using cross-validation. |
| **PySR** | | |
| binary operators | [+, -, *, /] | |
| unary operators | [exp, log, pow2, pow3] | Allowed unary operators. |
| maxsize | 50 | Maximum size of an expression tree. |
| ncycles per iteration | 380 | Cycles per iteration of the genetic algorithm. |
| parsimony | 0.001 | Penalty for complexity of expressions. |
| adaptive parsimony scaling | 1040 | Scales the penalty based on frequency of each complexity level |
| niterations for SR | 8000 | Iterations for SR. |
| num populations for SR | 15 | Number of populations for SR. |
| population size for SR | 30 | Number of expressions per population in SR |
| elementwise loss | MSE | Loss function used for SR. |
| batch size | 128 | batch size of the dataset used to evaluate loss of expressions |

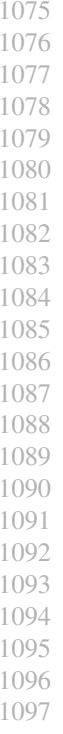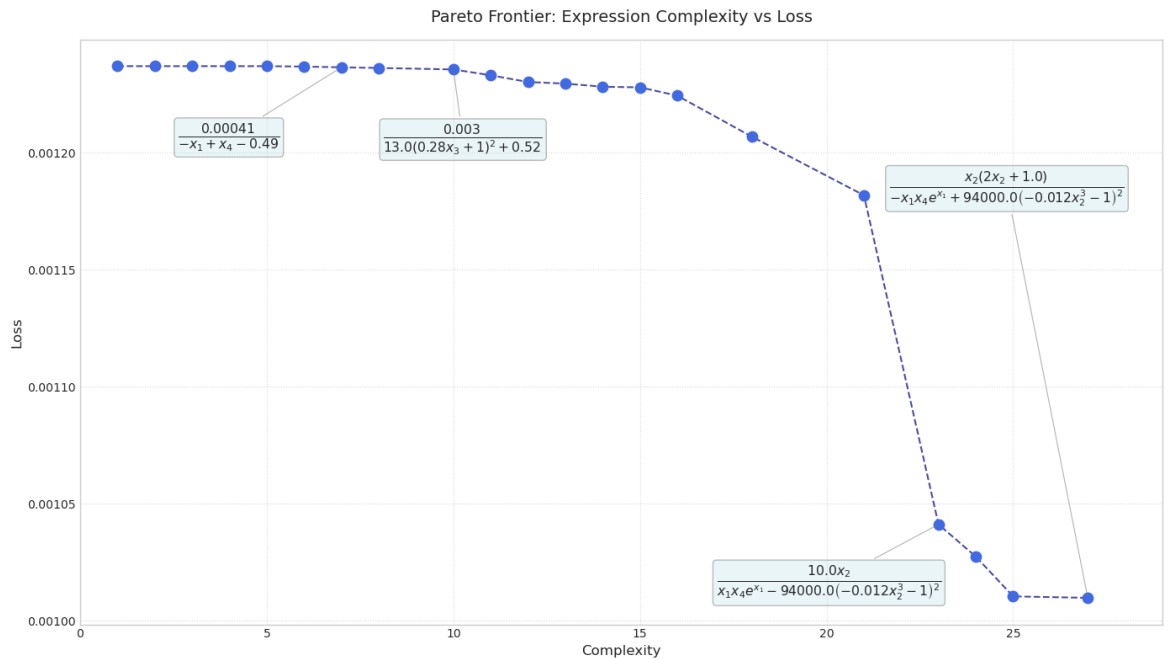

*Figure 15.* Pareto front of recovered expressions for the 4 dimensional Gaussian dataset.

*Table 10.* Expressions simplified with *sympy* alongside their raw complexity score from PySR for the Gaussian 4D dataset.

| Raw Complexity | Simplified Expression |
|:---:|:---|
| 1 | $4.0 \times 10^{-4}$ |
| 2 | $3.9 \times 10^{-4}$ |
| 3 | $4.0 \times 10^{-4}$ |
| 4 | $3.9 \times 10^{-4}$ |
| 5 | $4.0 \times 10^{-4}$ |
| 6 | $\dfrac{4.1 \times 10^{-4}}{e^{x_3} + 0.49}$ |
| 7 | $-\dfrac{4.1 \times 10^{-4}}{x_1 - x_4 + 0.49}$ |
| 8 | $0.003\,e^{-(x_3 + x_4)^2}$ |
| 10 | $\dfrac{0.003}{13\,(0.28\,x_3 + 1)^2 + 0.52}$ |
| 11 | $0$ |
| 12 | $0$ |
| 13 | $0$ |
| 14 | $0$ |
| 15 | $0$ |
| 16 | $-\dfrac{1.7 \times 10^{-6}}{x_3^3 \left(\dfrac{x_1}{x_3} + 0.87\right)^2}$ |
| 18 | $\dfrac{3.8 \times 10^{-9}}{(x_1 + 0.87\,x_3)^3}$ |
| 21 | $\dfrac{x_2}{x_1 x_4 e^{x_1} \ - \ 94000\left(-0.012\,x_2^3 - 1\right)^2}$ |
| 23 | $\dfrac{10\,x_2}{x_1 x_4 e^{x_1} \ - \ 94000\left(-0.012\,x_2^3 - 1\right)^2}$ |
| 24 | $\dfrac{9.3\,x_2}{x_1 x_4 e^{x_1} \ - \ 94000\left(-0.012\,x_2^3 - 1\right)^2}$ |
| 25 | $-\dfrac{x_2\,(x_2 - 3.6)}{x_1 x_4 e^{x_1} \ - \ 94000\left(-0.012\,x_2^3 - 1\right)^2}$ |
| 27 | $-\dfrac{x_2\,(2x_2 + 1)}{x_1 x_4 e^{x_1} \ - \ 94000\left(-0.012\,x_2^3 - 1\right)^2}$ |

*Table 11.* Hyper-parameters for symbolic regression on each independent component

| Hyper-parameter | Value | Notes |
|:---|:---|:---|
| **pgmpy Structure Learning** | | |
| CI test | Pearson correlation | Conditional independence test. |
| **Kernel Density Estimation** | | |
| number of samples | 225000 | |
| bandwidth | 0.0923 | KDE bandwidth selected using cross-validation. |
| **PySR** | | |
| binary operators | [+, -, *, /] | |
| unary operators | [exp, log, pow2, pow3] | Allowed unary operators. |
| maxsize | 50 | Maximum size of an expression tree. |
| ncycles per iteration | 380 | Cycles per iteration of the genetic algorithm. |
| parsimony | 0.001 | Penalty for complexity of expressions. |
| adaptive parsimony scaling | 1040 | Scales the penalty based on frequency of each complexity level |
| niterations for SR | 8000 | Iterations for SR. |
| num populations for SR | 15 | Number of populations for SR. |
| population size for SR | 30 | Number of expressions per population in SR |
| elementwise loss | MSE | Loss function used for SR. |
| batch size | 128 | batch size of the dataset used to evaluate loss of expressions |

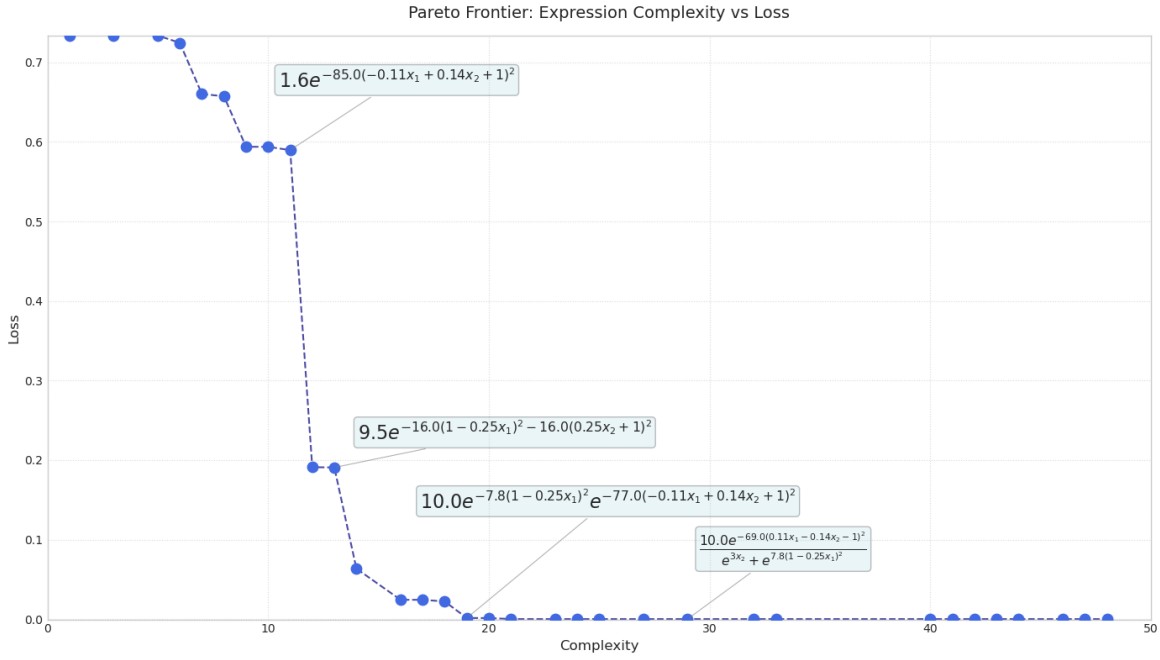

*Figure 16.* Pareto front of recovered expressions for the 4 dimensional Gaussian marginal $(x_1, x_2)$ dataset.

*Table 12.* Expressions simplified with *sympy* alongside their raw complexity score from PySR for the Gaussian 4D marginal $(x_1, x_2)$.

| Raw Complexity | Simplified Expression |
|---|---|
| 1 | $0.15$ |
| 3 | $0.15$ |
| 5 | $0.15$ |
| 6 | $\dfrac{x_1}{x_1^2 + 6.5}$ |
| 7 | $1.1\,e^{-16(0.25x_2+1)^2}$ |
| 8 | $\dfrac{1.1}{16\,(0.25x_2+1)^2 + 1.1}$ |
| 9 | $1.5\,e^{-64(-0.12x_1+0.12x_2+1)^2}$ |
| 10 | $1.4\,e^{-64(-0.12x_1+0.12x_2+1)^2}$ |
| 11 | $1.6\,e^{-85(-0.11x_1+0.14x_2+1)^2}$ |
| 12 | $9.9\,e^{-16(0.25x_1-1)^2-16(0.25x_2+1)^2}$ |
| 13 | $9.5\,e^{-16(1-0.25x_1)^2-16(0.25x_2+1)^2}$ |
| 14 | $11\,e^{-16(0.25x_2+1)^2-64(-0.12x_1+0.12x_2+1)^2}$ |
| 16 | $\dfrac{33\,e^{-64(-0.12x_1+0.12x_2+1)^2}}{(x_1+x_2)^2 + 3.2}$ |
| 17 | $\dfrac{33\,e^{-64(0.12x_1-0.12x_2-1)^2}}{(x_1+x_2)^2 + 3.2}$ |
| 18 | $\dfrac{30}{\left((x_1+x_2)^2 + 3.4\right)\left(e^{64(-0.12x_1+0.12x_2+1)^2} - 0.16\right)}$ |
| 19 | $10\,e^{-7.8(1-0.25x_1)^2-77(-0.11x_1+0.14x_2+1)^2}$ |
| 20 | $11\,e^{-7.8(1-0.25x_1)^2-77(-0.11x_1+0.14x_2+1)^2}$ |
| 21 | $10\,e^{-7.8(1-0.25x_1)^2-69(-0.11x_1+0.14x_2+1)^2}$ |
| 23 | $10\,e^{-7.8(1-0.25x_1)^2-68(-0.11x_1+0.14x_2+1)^2}$ |
| 24 | $10\,e^{-7.8(1-0.25x_1)^2-68(-0.11x_1+0.14x_2+1)^2}$ |
| 25 | $10\,e^{-7.8(1-0.25x_1)^2-69(0.11x_1-0.14x_2-1)^2}$ |
| 27 | $\left(10 - 6.9\times10^{-5}\,e^{70.84(1-0.25x_1)^2}\right)e^{-7.8(1-0.25x_1)^2-69(0.11x_1-0.14x_2-1)^2}$ |
| 29 | $\dfrac{10\,e^{-69(0.11x_1-0.14x_2-1)^2}}{e^{3x_2} + e^{70.84(1-0.25x_1)^2}}$ |
| 32 | $\dfrac{10\,e^{-69(0.11x_1-0.14x_2-1)^2}}{e^{70.84(1-0.25x_1)^2} + e^{-0.34x_1^3}}$ |
| 33 | $\dfrac{10\,e^{-69(0.11x_1-0.14x_2-1)^2}}{e^{70.84(1-0.25x_1)^2} + e^{-0.25x_1^3}}$ |
| 40 | $\left(-0.005\,x_2^3\,e^{-8.7(0.59x_1+0.59x_2-1)^2} + 9.9\right)e^{-7.8(-0.25x_2-1)^2-69(-0.14x_1+0.11x_2+1)^2}$ |
| 41 | $14\left(-3.3\times10^{-4}\,x_2^3\,e^{-8.7(0.59x_1+0.59x_2-1)^2} + 0.71\right)e^{-7.8(-0.25x_2-1)^2-69(-0.14x_1+0.11x_2+1)^2}$ |
| 42 | $\left(-0.005\,x_2^3\,e^{-8.7(0.59x_1+0.59x_2-1)^2} + 9.9\right)e^{-7.8(-0.25x_2-1)^2-69(-0.14x_1+0.11x_2+1)^2}$ |
| 43 | $1\left(-0.005\,x_2^3\,e^{-8.7(0.59x_1+0.59x_2-1)^2} + 9.9\right)e^{-7.8(-0.25x_2-1)^2-69(-0.14x_1+0.11x_2+1)^2}$ |
| 44 | $\left(-0.0034\,x_2^3\,e^{-8.7(0.59x_1+0.59x_2-1)^2} + 9.9\right)e^{-7.8(-0.25x_2-1)^2-69(-0.14x_1+0.11x_2+1)^2}$ |
| 46 | $\left(9.9 - 0.005\left(x_2\,e^{-2.9(0.59x_1+0.59x_2-1)^2} + 0.3\right)^3\right)e^{-7.8(-0.25x_2-1)^2-69(-0.14x_1+0.11x_2+1)^2}$ |
| 47 | $1\left(-0.005\,x_2^3\,e^{-12(0.65x_1+0.65x_2-1)^2} + 9.9\right)e^{-7.8(-0.25x_2-1)^2-69(-0.14x_1+0.11x_2+1)^2}$ |
| 48 | $\left(9.9 - 0.005\left(x_2\,e^{-2.9(0.59x_1+0.59x_2-1)^2} + 0.3\right)^3\right)e^{-7.8(-0.25x_2-1)^2-69(-0.14x_1+0.11x_2+1)^2}$ |

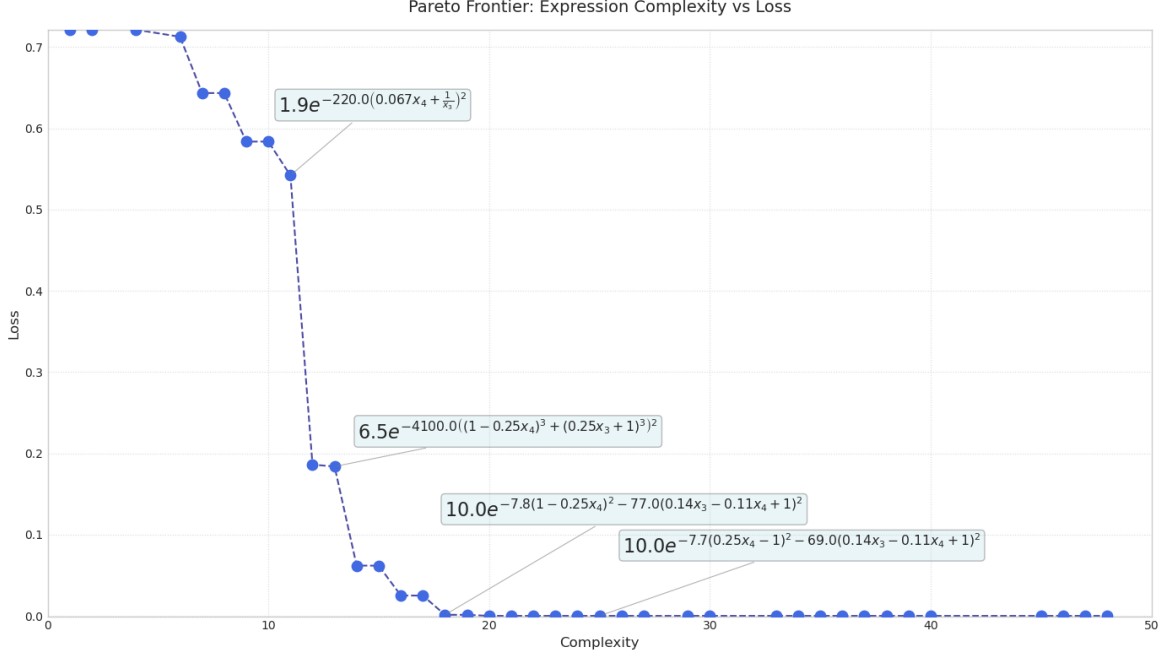

*Figure 17.* Pareto front of recovered expressions for the 4 dimensional Gaussian marginal $(x_3, x_4)$ dataset.

*Table 13.* Expressions simplified with *sympy* alongside their raw complexity score from PySR for the Gaussian 4D Marginal $(x_3, x_4)$ dataset.

| Raw Complexity | Simplified Expression |
|---|---|
| 1 | $0.15$ |
| 2 | $0.15$ |
| 4 | $0.14$ |
| 6 | $\dfrac{0.1}{e^{x_3} + 0.49}$ |
| 7 | $e^{-8.3(1-0.25x_4)^2}$ |
| 8 | $e^{-8.6(0.25x_4-1)^2}$ |
| 9 | $1.5e^{-64(0.12x_3-0.12x_4+1)^2}$ |
| 10 | $1.5e^{-64(0.12x_3-0.12x_4+1)^2}$ |
| 11 | $1.9e^{-220\left(0.067x_4+\frac{1}{x_3}\right)^2}$ |
| 12 | $9.4e^{-16(1-0.25x_4)^2-16(0.25x_3+1)^2}$ |
| 13 | $6.5e^{-4100\left((1-0.25x_4)^3+(0.25x_3+1)^3\right)^2}$ |
| 14 | $10e^{-16(0.25x_4-1)^2-64(0.12x_3-0.12x_4+1)^2}$ |
| 15 | $9.5e^{-16(0.25x_4-1)^2-64(0.12x_3-0.12x_4+1)^2}$ |
| 16 | $9.3e^{-8.4(1-0.25x_4)^2-64(0.12x_3-0.12x_4+1)^2}$ |
| 17 | $9.3e^{-8.4(0.25x_4-1)^2-64(-0.12x_3+0.12x_4-1)^2}$ |
| 18 | $10e^{-7.8(1-0.25x_4)^2-77(0.14x_3-0.11x_4+1)^2}$ |
| 19 | $9.5e^{-7.8(0.25x_4-1)^2-77(0.14x_3-0.11x_4+1)^2}$ |
| 20 | $9.9e^{-7.8(0.25x_4-1)^2-71(0.13x_3-0.11x_4+1)^2}$ |
| 21 | $9.9e^{-7.8(0.25x_4-1)^2-71(0.13x_3-0.11x_4+1)^2}$ |
| 22 | $9.9e^{-7.7(1-0.25x_4)^2-69(0.14x_3-0.11x_4+1)^2}$ |
| 23 | $10e^{-7.7(0.25x_4-1)^2-69(0.14x_3-0.11x_4+1)^2}$ |
| 24 | $10e^{-7.7(1-0.25x_4)^2-69(0.14x_3-0.11x_4+1)^2}$ |
| 25 | $10e^{-7.7(0.25x_4-1)^2-69(0.14x_3-0.11x_4+1)^2}$ |
| 26 | $9.9e^{-7.7(1-0.25x_4)^2-69(0.14x_3-0.11x_4+1)^2}$ |
| 27 | $9.9e^{-7.7(0.25x_4-1)^2-69(0.14x_3-0.11x_4+1)^2}$ |
| 29 | $(9.9e^{x_4}+0.35)\,e^{-x_4-7.7(1-0.25x_4)^2-69(0.14x_3-0.11x_4+1)^2}$ |
| 30 | $(-x_3+9.9e^{2x_4})\,e^{-2x_4}e^{-7.7(1-0.25x_4)^2-69(0.14x_3-0.11x_4+1)^2}$ |
| 33 | $(-x_3e^{-2x_4}+9.9)\,e^{-7.7(1-0.25x_4)^2-69(-0.14x_3+0.11x_4-1)^2}$ |
| 34 | $\left(e^{-9.6x_3^2(0.32x_3+1)^2+x_3}+9.9\right)e^{-7.7(0.25x_4-1)^2-69(0.14x_3-0.11x_4+1)^2}$ |
| 35 | $\left(9.9e^{90.61x_3^2(0.32x_3+1)^2}+0.058\right)e^{-9.6x_3^2(0.32x_3+1)^2-7.7(0.25x_4-1)^2-69(0.14x_3-0.11x_4+1)^2}$ |
| 36 | $\left(9.9e^{90.61x_3^2(0.32x_3+1)^2+0.56x_4}+1\right)e^{-9.6x_3^2(0.32x_3+1)^2-0.56x_4-7.7(0.25x_4-1)^2-69(0.14x_3-0.11x_4+1)^2}$ |
| 37 | $\left(9.9e^{98\left(1-0.057x_3^2\right)^2}+0.1\right)e^{-98\left(1-0.057x_3^2\right)^2-7.7(0.25x_4-1)^2-69(0.14x_3-0.11x_4+1)^2}$ |
| 38 | $\left(9.9e^{41.8\left(1-0.057x_3^2\right)^2}+0.1\right)e^{-400\left(1-0.057x_3^2\right)^2-7.7(0.25x_4-1)^2-69(-0.14x_3+0.11x_4-1)^2}$ |
| 39 | $0.13\left(75e^{41.8\left(1-0.057x_3^2\right)^2}+1\right)e^{-400\left(1-0.057x_3^2\right)^2-7.7(0.25x_4-1)^2-69(-0.14x_3+0.11x_4-1)^2}$ |
| 40 | $\left(9.9e^{x_4\left(1500x_4\left(0.016-(-0.29x_3-1)^3\right)^2+0.56\right)}+1\right)e^{-x_4\left(1500x_4\left(0.016-(-0.29x_3-1)^3\right)^2+0.56\right)-7.7(0.25x_4-1)^2-69(0.14x_3-0.11x_4+1)^2}$ |
| 45 | $\left(9.9e^{x_4\left(1100x_4\left(-0.031x_3-(-0.31x_3-1)^3-0.098\right)^2+0.56\right)}+1\right)e^{-x_4\left(1100x_4\left(-0.031x_3-(-0.31x_3-1)^3-0.098\right)^2+0.56\right)-7.7(0.25x_4-1)^2-69(0.14x_3-0.11x_4+1)^2}$ |
| 46 | $0.16\left(61e^{3480x_4^2\left(0.031x_3+(-0.31x_3-1)^3+0.098\right)^2}+1\right)e^{-3500x_4^2\left(0.031x_3+(-0.31x_3-1)^3+0.098\right)^2-7.7(0.25x_4-1)^2-69(0.14x_3-0.11x_4+1)^2}$ |
| 47 | $\left(9.9e^{x_4\left(2100x_4\left(-0.031x_3-(-0.31x_3-1)^3-0.098\right)^2+0.56\right)}+1\right)e^{-x_4\left(2100x_4\left(-0.031x_3-(-0.31x_3-1)^3-0.098\right)^2+0.56\right)-7.7(0.25x_4-1)^2-69(-0.14x_3+0.11x_4-1)^2}$ |
| 48 | $0.16\left(61e^{3480x_4^2\left(0.031x_3+(-0.31x_3-1)^3+0.098\right)^2}+1\right)e^{-3500x_4^2\left(0.031x_3+(-0.31x_3-1)^3+0.098\right)^2-7.7(0.25x_4-1)^2-69(0.14x_3-0.11x_4+1)^2}$ |

## C.3. Rastrigin Dataset

*Table 14.* Hyper-parameters for symbolic regression on the Rastrigin dataset

| Hyper-parameter | Value | Notes |
|---|---|---|
| **Kernel Density Estimation** | | |
| number of samples | 900000 | |
| bandwidth | 0.0254 | KDE bandwidth selected using cross-validation. |
| **PySR** | | |
| binary operators | [+, -, *, /] | |
| unary operators | [exp, log, pow2, cos, sin] | Allowed unary operators. |
| maxsize | 50 | Maximum size of an expression tree. |
| ncycles per iteration | 380 | Cycles per iteration of the genetic algorithm. |
| parsimony | 0.001 | Penalty for complexity of expressions. |
| adaptive parsimony scaling | 1040 | Scales the penalty based on frequency of each complexity level |
| niterations for SR | 8000 | Iterations for SR. |
| num populations for SR | 15 | Number of populations for SR. |
| population size for SR | 30 | Number of expressions per population in SR |
| elementwise loss | MSE | Loss function used for SR. |
| batch size | 128 | batch size of the dataset used to evaluate loss of expressions |

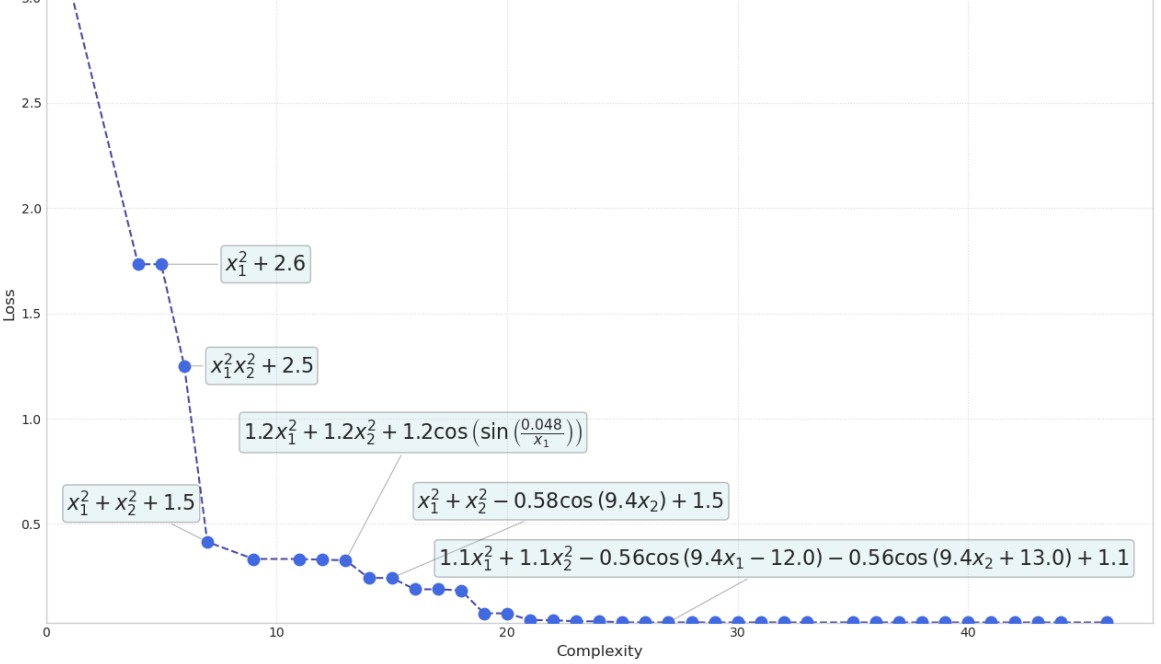

*Figure 18.* Pareto front of recovered expressions for the Rastrigin dataset.

*Table 15.* Expressions simplified with *sympy* alongside their raw complexity score from PySR for the Rastrigin dataset.

| Raw Complexity | Simplified Expression |
|---|---|
| 1 | $3.6$ |
| 4 | $x_1^2 + 2.6$ |
| 5 | $x_1^2 + 2.6$ |
| 6 | $x_1^2 x_2^2 + 2.5$ |
| 7 | $x_1^2 + x_2^2 + 1.5$ |
| 9 | $1.2x_1^2 + 1.2x_2^2 + 1$ |
| 11 | $1.2x_1^2 + 1.2x_2^2 + 1$ |
| 12 | $1.7x_1^2 + 1.2x_2^2 + \cos(x_1)$ |
| 13 | $1.2x_1^2 + 1.2x_2^2 + 1.2\cos\left(\sin\left(\frac{0.048}{x_1}\right)\right)$ |
| 14 | $x_1^2 + x_2^2 - 0.58\cos(9.4x_2) + 1.5$ |
| 15 | $x_1^2 + x_2^2 - 0.58\cos(9.4x_2) + 1.5$ |
| 16 | $1.2x_1^2 + 1.2x_2^2 - 0.55\cos(9.4x_2) + 1.1$ |
| 17 | $1.2x_1^2 + 1.2x_2^2 - 0.55\cos(9.4x_2) + 1.1$ |
| 18 | $1.2x_1^2 + 1.2x_2^2 - 0.55\cos(9.4x_2 + 13) + 1.1$ |
| 19 | $x_1^2 + x_2^2 - 0.58\cos(9.4x_1) - 0.58\cos(9.4x_2) + 1.4$ |
| 20 | $x_1^2 + x_2^2 - 0.58\cos(9.4x_1) - 0.58\cos(9.4x_2) + 1.4$ |
| 21 | $1.1x_1^2 + 1.1x_2^2 - 0.55\cos(9.4x_1) - 0.55\cos(9.4x_2) + 1.1$ |
| 22 | $1.1x_1^2 + 1.1x_2^2 - 0.55\cos(9.4x_1) - 0.55\cos(9.4x_2) + 1.1$ |
| 23 | $1.1x_1^2 + 1.1x_2^2 - 0.55\cos(9.4x_2) - 0.55\cos(9.4x_1 + 6.5) + 1.1$ |
| 24 | $1.1x_1^2 + 1.1x_2^2 - 0.55\cos(9.4x_2) - 0.55\cos(9.4x_1 + 6.5) + 1.1$ |
| 25 | $1.1x_1^2 + 1.1x_2^2 - 0.56\cos(9.4x_1 - 12) - 0.56\cos(9.4x_2 + 13) + 1.1$ |
| 26 | $1.1x_1^2 + 1.1x_2^2 - 0.56\cos(9.4x_1 - 12) - 0.56\cos(9.4x_2 - 12) + 1.1$ |
| 27 | $1.1x_1^2 + 1.1x_2^2 - 0.56\cos(9.4x_1 - 12) - 0.56\cos(9.4x_2 + 13) + 1.1$ |
| 28 | $1.1x_1^2 + 1.1x_2^2 + 1.1\cos(0.098x_2) - 0.56\cos(9.4x_1 - 12) - 0.56\cos(9.4x_2 + 13)$ |
| 29 | $1.1x_1^2 + 1.1x_2^2 - 0.56\cos(9.4x_1 - 12) - 0.56\cos(9.4x_2 + 0.18) + 1.1\cos(0.16\sin(x_2))$ |
| 30 | $1.1x_2^2 + 1.1(0.0022 - x_1)^2 + 1.1\cos(0.098x_2) - 0.56\cos(9.4x_1 - 12) - 0.56\cos(9.4x_2 + 13)$ |
| 31 | $1.1x_2^2 + 1.1(x_1 - 0.002)^2 - 0.56\cos(9.4x_1 - 12) - 0.56\cos(9.4x_2 - 12) + 1.1\cos(0.15\sin(x_2))$ |
| 32 | $1.1x_2^2 + 1.1(x_1 - 0.002)^2 - 0.56\cos(9.4x_1 - 12) - 0.56\cos(9.4x_2 - 12) + 1.1\cos(0.15\sin(x_2))$ |
| 33 | $1.1x_2^2 + 1.1(0.002 - x_1)^2 - 0.56\cos(9.4x_1 - 12) - 0.56\cos(9.4x_2 - 12) + 1.1\cos(0.18\sin(x_2)) + 0.0028$ |
| 34 | $1.1x_2^2 + 1.1(x_1 - 0.0028)^2 - 0.55\cos(9.4x_1 - 12) - 0.55\cos(9.4x_2 - 12) + 1.1\cos(0.26\sin(x_2))$ |
| 35 | $1.1x_1^2 + 1.1x_2^2 - 0.57\cos(0.18x_1x_2)\cos(9.4x_2 + 13) - 0.55\cos(9.4x_1 - 12) + 1.1$ |
| 36 | $1.1x_1^2 + 1.1x_2^2 - 0.57\cos(0.18x_1x_2)\cos(9.4x_2 - 12) - 0.55\cos(9.4x_1 - 6.1) + 1.1$ |
| 37 | $1.1x_1^2 + 1.1x_2^2 - 0.56\cos(0.16x_1x_2)\cos(9.4x_2 - 12) - 0.55\cos(9.4x_1 - 12) + 1.1\cos(0.25\sin(x_2))$ |
| 38 | $1.1x_1^2 + 1.1x_2^2 - 0.56\cos(0.17x_1x_2)\cos(9.4x_2 + 13) - 0.55\cos(9.4x_1 - 12) + 1.1\cos(0.25\sin(x_2))$ |
| 39 | $1.1x_2^2 + 1.1(x_1 - 0.00076)^2 - 0.56\cos(0.16x_1x_2)\cos(9.4x_2 - 12) - 0.55\cos(9.4x_1 - 12) + 1.1\cos(0.25\sin(x_2))$ |
| 40 | $1.1x_2^2 + 1.1(x_1 - 0.0019)^2 - 0.56\cos(0.17x_1x_2)\cos(9.4x_2 - 12) - 0.55\cos(9.4x_1 - 12) + 1.1\cos(0.25\sin(x_2)) - 0.002$ |

## C.4. Muon Decay Dataset

For consistency with other examples we denote $m_{13}^2$ as $x_1$ and $m_{23}^2$ as $x_2$.

*Table 16.* Hyper-parameters for symbolic regression on the Muon Decay dataset

| Hyper-parameter | Value | Notes |
|---|---|---|
| | **Kernel Density Estimation** | |
| number of samples | 450000 | |
| bandwidth | 0.0114 | KDE bandwidth selected using cross-validation. |
| | **PySR** | |
| binary operators | [+, -, *, /] | |
| unary operators | [exp, log, pow2, pow3] | Allowed unary operators. |
| maxsize | 30 | Maximum size of an expression tree. |
| ncycles per iteration | 380 | Cycles per iteration of the genetic algorithm. |
| parsimony | 0.001 | Penalty for complexity of expressions. |
| adaptive parsimony scaling | 1040 | Scales the penalty based on frequency of each complexity level |
| niterations for SR | 8000 | Iterations for SR. |
| num populations for SR | 15 | Number of populations for SR. |
| population size for SR | 30 | Number of expressions per population in SR |
| elementwise loss | MSE | Loss function used for SR. |
| batch size | 128 | batch size of the dataset used to evaluate loss of expressions |

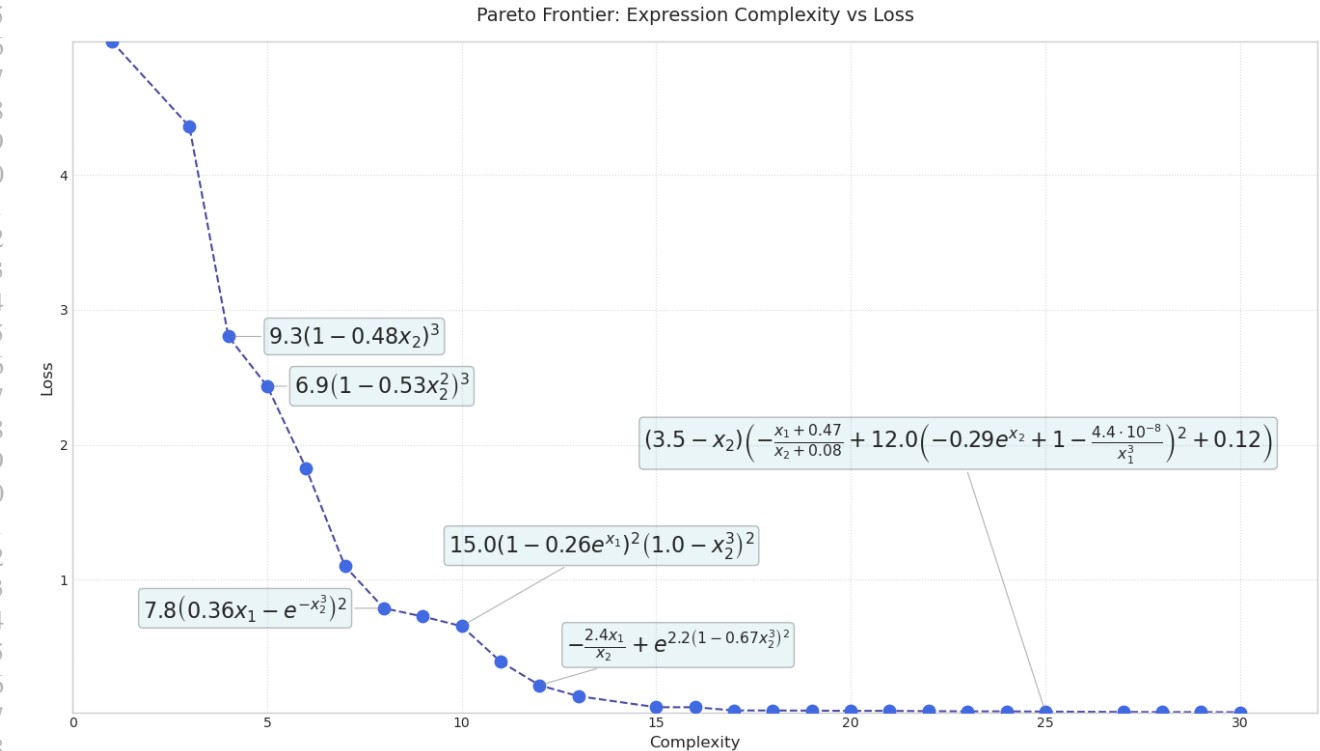

*Figure 19.* Pareto front of recovered expressions for the muon decay dataset.

*Table 17.* Expressions simplified with *sympy* alongside their raw complexity score from PySR for the Muon decay dataset.

| Raw Complexity | Simplified Expression |
| --- | --- |
| 1 | $4$ |
| 3 | $4.5 - x_2$ |
| 4 | $9.3 \left(1 - 0.48 x_2\right)^3$ |
| 5 | $6.9 \left(1 - 0.53 x_2^2\right)^3$ |
| 6 | $8.4 \left(0.34 x_1 + 0.34 x_2 - 1\right)^2$ |
| 7 | $8.6 - \left(x_1 + e^{x_2}\right)^2$ |
| 8 | $7.8 \left(0.36 x_1 - e^{-x_2^3}\right)^2$ |
| 9 | $8.3 \left(0.42 x_1 - 1\right)^2 \left(0.83 x_2^3 - 1\right)^2$ |
| 10 | $15 \left(x_2^3 - 1\right)^2 \left(0.26 e^{x_1} - 1\right)^2$ |
| 11 | $-\dfrac{2.3 x_1}{x_2} + 25 \left(0.3 e^{x_2} - 1\right)^2$ |
| 12 | $-\dfrac{2.4 x_1}{x_2} + e^{20.25\left(1 - 0.67 x_2^3\right)^2}$ |
| 13 | $\dfrac{-2.5 x_1 + 27 x_2 \left(0.3 e^{x_2} - 1\right)^2 - 0.21}{x_2}$ |
| 15 | $\dfrac{-3.1 x_1 + 36 \left(1 - 0.29 e^{x_2}\right)^2 \left(x_2 + 0.071\right) - 1.2}{x_2 + 0.071}$ |
| 16 | $\dfrac{3.1 \left(-x_1 + 12 \left(1 - 0.29 e^{x_2}\right)^2 \left(x_2 + 0.071\right) - 0.39\right)}{x_2 + 0.071}$ |
| 17 | $\dfrac{\left(x_2 - 3.3\right) \left(x_1 - 12 \left(x_2 + 0.069\right) \left(0.29 e^{x_2} - 1\right)^2 + 0.44\right)}{x_2 + 0.069}$ |
| 18 | $\dfrac{\left(x_2 - 3.4\right) \left(x_1 - 12 \left(1 - 0.29 e^{x_2}\right)^2 \left(x_2 + 0.069\right) + 0.44\right)}{x_2 + 0.069}$ |
| 19 | $\dfrac{\left(x_2 - 3.4\right) \left(x_1 - \left(x_2 + 0.077\right) \left(12 \left(0.29 e^{x_2} - 1\right)^2 + 0.11\right) + 0.47\right)}{x_2 + 0.077}$ |
| 20 | $\dfrac{\left(e^{x_2} - 4.7\right) \left(x_1 - \left(x_2 + 0.097\right) \left(11 \left(1 - 0.3 e^{x_2}\right)^2 + 0.38\right) + 0.56\right)}{x_2 + 0.097}$ |
| 21 | $\left(0.13 x_1 + \log\left(x_2\right)\right) \left(6.9 x_1 + 16 x_2 \left(\log\left(x_2\right) - 0.68\right) + 1.2\right)$ |
| 22 | $-x_1 + \left(-x_1^2 + 7.1 x_1 + 17 x_2 \left(\log\left(x_2\right) - 0.66\right) + 1.1\right) \log\left(x_2\right)$ |
| 23 | $\dfrac{\left(x_2 - 3.4\right) \left(x_1 - 12 \left(x_2 + 0.069\right) \left(0.29 e^{x_2} - 1 + \frac{4.6 \cdot 10^{-8}}{x_1^3}\right)^2 + 0.44\right)}{x_2 + 0.069}$ |
| 24 | $\dfrac{\left(x_2 - 3.4\right) \left(x_1 + 0.44 - \frac{\left(x_2 + 0.069\right)\left(x_1^3 \left(e^{x_2} - 3.4\right) + 1.6 \cdot 10^{-7}\right)^2}{x_1^6}\right)}{x_2 + 0.069}$ |
| 25 | $\dfrac{\left(x_2 - 3.5\right) \left(x_1 - \left(x_2 + 0.08\right) \left(12 \left(-0.29 e^{x_2} + 1 - \frac{4.4 \cdot 10^{-8}}{x_1^3}\right)^2 + 0.12\right) + 0.47\right)}{x_2 + 0.08}$ |
| 27 | $-x_1 + \left(-x_1^2 + 7.2 x_1 + 17 x_2 \left(\log\left(x_2\right) - 0.66\right) + 1.1\right) \log\left(x_2\right) - \frac{1.3 \cdot 10^{-6}}{x_1^3}$ |
| 28 | $-x_1 + 17 \left(-0.059 x_1^2 + 0.44 x_1 - x_2 \left(0.66 - \log\left(x_2\right)\right) + 0.065\right) \log\left(x_2 + \frac{6.9 \cdot 10^{-8}}{x_1^3}\right)$ |
| 29 | $-x_1 + 17 \left(-0.059 x_1^3 + 0.42 x_1 - x_2 \left(0.65 - \log\left(x_2\right)\right) + 0.065\right) \log\left(x_2 + \frac{7.5 \cdot 10^{-8}}{x_1^3}\right)$ |
| 30 | $-x_1 + 17 \left(-0.1 x_1^2 + 0.47 x_1 - x_2 \left(0.66 - \log\left(x_2\right)\right) + 0.065\right) \log\left(x_2 + \frac{8 \cdot 10^{-8}}{x_1^3}\right)$ |

## C.5. Dijet Dataset

*Table 18.* Hyper-parameters for symbolic regression on the dijet dataset (using loss SR MSE + NLL + NP)

| Hyper-parameter | Value | Notes |
|---|---|---|
| **Neural Spline Flow** | | |
| K | 16 | Number of autoregressive rational quadratic spline flow blocks. |
| hidden units | 128 | Width of the neural networks inside each autoregressive spline flow. |
| hidden layers | 2 | Depth of the neural networks inside each autoregressive spline flow |
| number of samples | 90000 | |
| epochs | 1390 | Number of training epochs |
| batch size | 5000 | batch size of samples used to train flow |
| learning rate | 3e-5 | |
| weight decay | 1e-5 | weight decay regularization |
| **PySR** | | |
| binary operators | [+, -, *, /] | |
| unary operators | [exp, log, pow2, pow3] | Allowed unary operators. |
| maxsize | 30 | Maximum size of an expression tree. |
| ncycles per iteration | 380 | Cycles per iteration of the genetic algorithm. |
| parsimony | 0.001 | Penalty for complexity of expressions. |
| adaptive parsimony scaling | 1040 | Scales the penalty based on frequency of each complexity level |
| niterations for SR | 8000 | Iterations for SR. |
| num populations for SR | 15 | Number of populations for SR. |
| population size for SR | 30 | Number of expressions per population in SR |
| elementwise loss | MSE + NLL + NP | Loss function used for SR. |
| batch size | 1000 | batch size of the dataset used to evaluate loss of expressions |

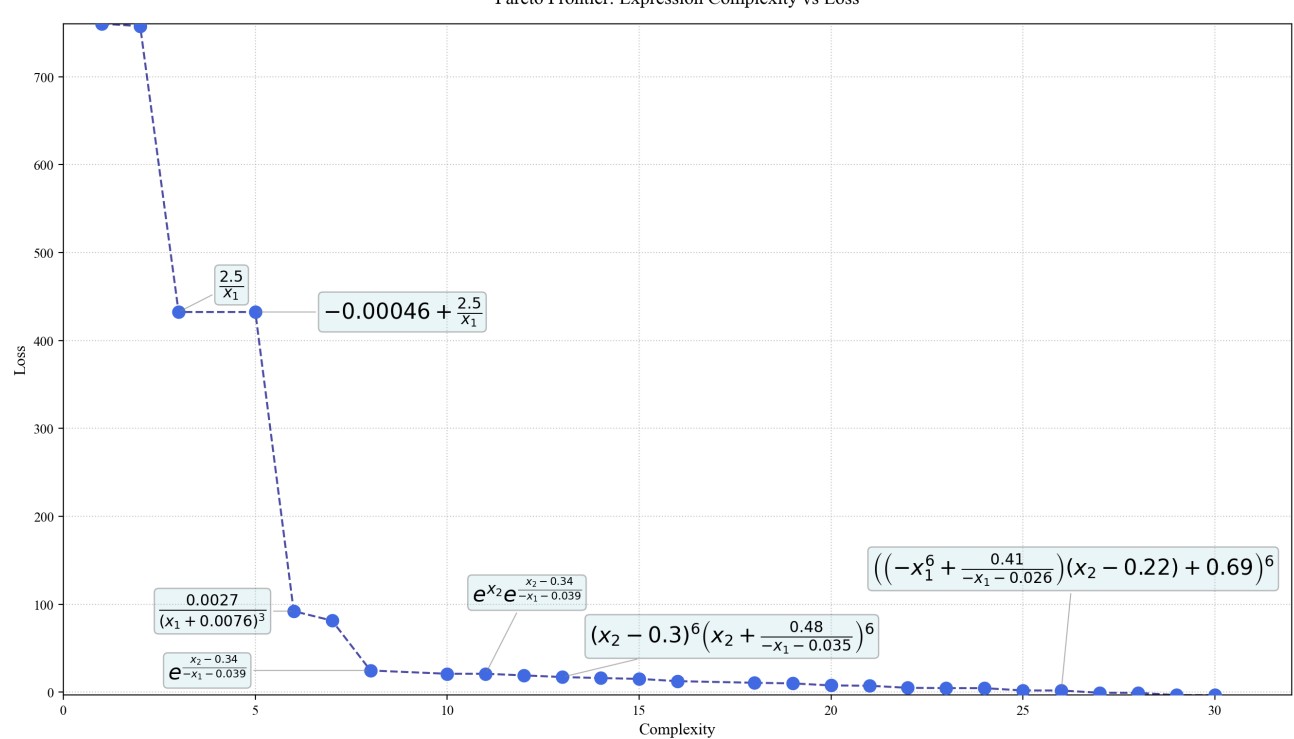

*Figure 20.* Pareto front of recovered expressions for the dijet dataset (SR MSE + NLL + NP).

*Table 19.* Expressions simplified with *sympy* alongside their raw complexity score from PySR for the dijet dataset (SR MSE + NLL + NP).

| Raw Complexity | Simplified Expression |
|---|---|
| 1 | $0.88$ |
| 2 | $e^{x_2}$ |
| 3 | $\dfrac{2.5}{x_1}$ |
| 5 | $-4.6 \times 10^{-4} + \dfrac{2.5}{x_1}$ |
| 6 | $\dfrac{0.0027}{(x_1 + 0.0076)^3}$ |
| 7 | $e^{\frac{0.029}{(x_1 + 0.058)^2}}$ |
| 8 | $e^{\frac{0.34 - x_2}{x_1 + 0.039}}$ |
| 10 | $e^{\frac{x_2(x_1 + 0.039) - x_2 + 0.34}{x_1 + 0.039}}$ |
| 11 | $e^{x_2 + \frac{0.34 - x_2}{x_1 + 0.039}}$ |
| 12 | $e^{\frac{x_2(x_1 + 0.039) - x_2 + 0.34}{x_1 + 0.039}} - 1.3$ |
| 13 | $\dfrac{(x_2 - 0.3)^6 \left(x_2(x_1 + 0.035) - 0.48\right)^6}{(x_1 + 0.035)^6}$ |
| 14 | $\dfrac{x_1 e^{\frac{x_2(x_1 + 0.039) - x_2 + 0.34}{x_1 + 0.039}} - 0.23}{x_1}$ |
| 15 | $\dfrac{(x_2 - 0.3)^6 \left(x_2^2(x_1 + 0.035) - 0.48\right)^6}{(x_1 + 0.035)^6}$ |
| 16 | $\dfrac{(x_2 - 0.3)^6 \left(x_2(x_1 + 0.035) - 0.48\right)^6 e^{x_2}}{(x_1 + 0.035)^6}$ |
| 18 | $\dfrac{(x_2 - 0.29)^6 \left(x_2^2(x_1 + 0.035) - 0.5\right)^6 e^{x_2}}{(x_1 + 0.035)^6}$ |
| 19 | $\dfrac{\left(0.68 x_1 - (x_2 - 0.22)\left(x_1^3(x_1 + 0.026) + 0.41\right) + 0.018\right)^6}{(x_1 + 0.026)^6}$ |
| 20 | $\dfrac{\left(0.71 x_1 - (x_2 - 0.21)\left(x_1^2(x_1 + 0.026) + 0.43\right) + 0.018\right)^6 e^{x_2}}{(x_1 + 0.026)^6}$ |
| 21 | $\dfrac{\left(0.68 x_1 - (x_2 - 0.22)\left(x_1^4(x_1 + 0.026) + 0.41\right) + 0.018\right)^6}{(x_1 + 0.026)^6}$ |
| 22 | $\dfrac{\left(0.71 x_1 - (x_2 - 0.21)\left(x_1^3(x_1 + 0.026) + 0.43\right) + 0.018\right)^6 e^{x_2}}{(x_1 + 0.026)^6}$ |
| 23 | $\dfrac{\left(0.68 x_1 - (x_2 - 0.22)\left(x_1^5(x_1 + 0.026) + 0.41\right) + 0.018\right)^6}{(x_1 + 0.026)^6}$ |
| 24 | $\dfrac{\left(0.71 x_1 - (x_2 - 0.21)\left((x_1 + 0.026)(x_1^3 - x_2) + 0.43\right) + 0.018\right)^6 e^{x_2}}{(x_1 + 0.026)^6}$ |
| 25 | $\dfrac{\left(0.68 x_1 - (x_2 - 0.22)\left(x_1^6(x_1 + 0.026) + 0.41\right) + 0.018\right)^6}{(x_1 + 0.026)^6}$ |
| 26 | $\dfrac{\left(0.69 x_1 - (x_2 - 0.22)\left(x_1^6(x_1 + 0.026) + 0.41\right) + 0.018\right)^6}{(x_1 + 0.026)^6}$ |
| 27 | $\dfrac{\left(0.68 x_1 - (x_2 - 0.22)\left(x_1^7(x_1 + 0.026) + 0.41\right) + 0.018\right)^6}{(x_1 + 0.026)^6}$ |
| 28 | $\dfrac{\left(0.69 x_1 - (x_2 - 0.22)\left(x_1^7(x_1 + 0.026) + 0.41\right) + 0.018\right)^6}{(x_1 + 0.026)^6}$ |
| 29 | $\dfrac{\left(0.68 x_1 - (x_2 - 0.22)\left(x_1^8(x_1 + 0.026) + 0.41\right) + 0.018\right)^6}{(x_1 + 0.026)^6}$ |
| 30 | $\dfrac{\left(0.69 x_1 - (x_2 - 0.22)\left(x_1^8(x_1 + 0.026) + 0.41\right) + 0.018\right)^6}{(x_1 + 0.026)^6}$ |

*Table 20.* Hyper-parameters for symbolic regression on the dijet dataset (SR MSE + NP)

| Hyper-parameter | Value | Notes |
|---|---|---|
| **Neural Spline Flow** | | |
| K | 16 | Number of autoregressive rational quadratic spline flow blocks. |
| hidden units | 128 | Width of the neural networks inside each autoregressive spline flow. |
| hidden layers | 2 | Depth of the neural networks inside each autoregressive spline flow |
| number of samples | 90000 | |
| epochs | 1390 | Number of training epochs |
| batch size | 5000 | batch size of samples used to train flow |
| learning rate | 3e-5 | |
| weight decay | 1e-5 | weight decay regularization |
| **PySR** | | |
| binary operators | [+, -, *, /] | |
| unary operators | [exp, log, pow2, pow3] | Allowed unary operators. |
| maxsize | 30 | Maximum size of an expression tree. |
| ncycles per iteration | 380 | Cycles per iteration of the genetic algorithm. |
| parsimony | 0.001 | Penalty for complexity of expressions. |
| adaptive parsimony scaling | 1040 | Scales the penalty based on frequency of each complexity level |
| niterations for SR | 8000 | Iterations for SR. |
| num populations for SR | 15 | Number of populations for SR. |
| population size for SR | 30 | Number of expressions per population in SR |
| elementwise loss | MSE + NP | Loss function used for SR. |
| batch size | 1000 | batch size of the dataset used to evaluate loss of expressions |

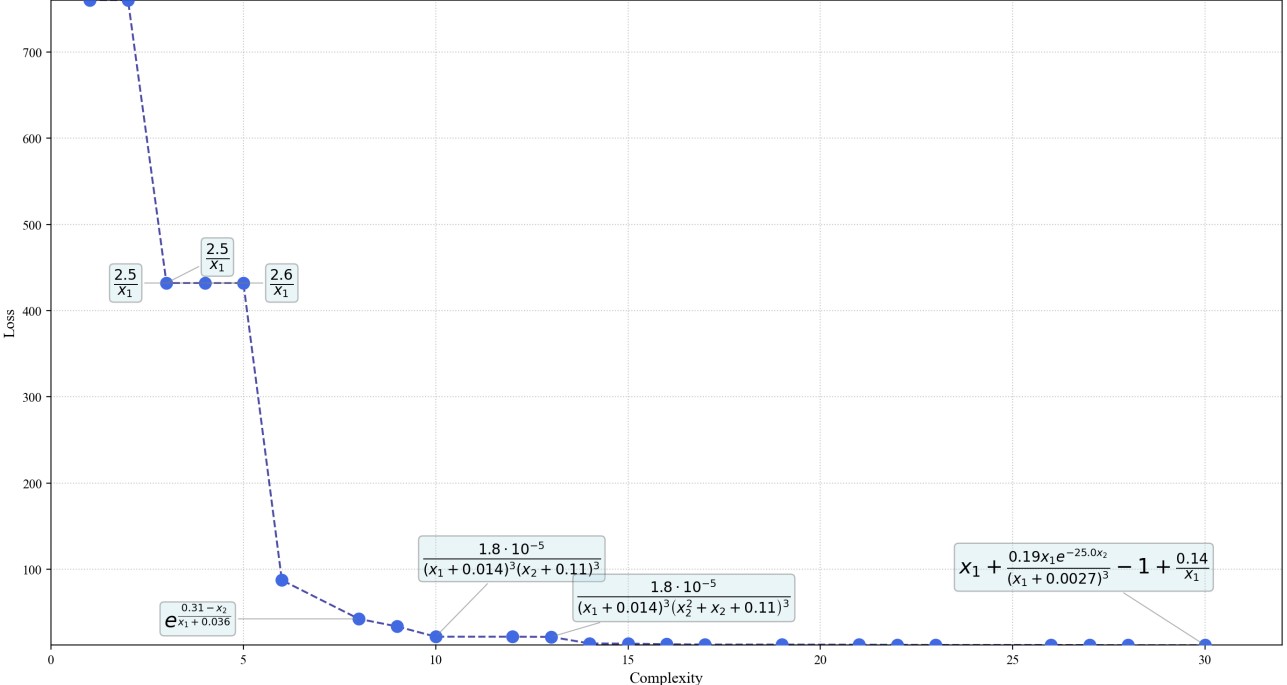

*Figure 21.* Pareto front of recovered expressions for the dijet dataset (SR MSE + NP).

*Table 21.* Expressions simplified with *sympy* alongside their raw complexity score for the dijet dataset recovered using SR MSE + NP.

| Raw Complexity | Simplified Expression |
|---|---|
| 1 | $0.82$ |
| 2 | $0.82$ |
| 3 | $\dfrac{2.5}{x_1}$ |
| 4 | $\dfrac{2.6}{x_1}$ |
| 5 | $\dfrac{2.5}{x_1}$ |
| 6 | $\dfrac{0.0027}{(x_1 + 0.008)^3}$ |
| 8 | $e^{\frac{0.31 - x_2}{x_1}} + 0.036$ |
| 9 | $e^{\frac{2(0.23 - x_2)}{x_1}} + 0.054$ |
| 10 | $\dfrac{1.8 \times 10^{-5}}{(x_1 + 0.014)^3 (x_2 + 0.11)^3}$ |
| 12 | $\dfrac{1.8 \times 10^{-5}}{(x_1 + 0.014)^3 (x_2 + 0.11)^3}$ |
| 13 | $\dfrac{1.8 \times 10^{-5}}{(x_1 + 0.014)^3 (x_2^2 + x_2 + 0.11)^3}$ |
| 14 | $\dfrac{1.5 \times 10^{-4} x_1}{(x_1 + 0.0028)^3 (x_2 + 0.086)^3}$ |
| 15 | $\dfrac{1.5 \times 10^{-4} x_1}{(x_1 + 0.0027)^3 (x_2 + 0.09)^3}$ |
| 16 | $\dfrac{2.6 \times 10^{-4} x_1}{(x_1 + 0.0027)^3 (x_2 + 0.11)^3}$ |
| 17 | $\dfrac{2.2 \times 10^{-4} x_1}{(x_1 + 0.0027)^3 (x_2 + 0.1)^3}$ |
| 19 | $\dfrac{1.3 \times 10^{-4} x_1 (x_2 + e^{x_2})^3}{(x_1 + 0.0027)^3 (x_2 (x_2 + e^{x_2}) + 0.086)^3}$ |
| 21 | $\dfrac{0.19 x_1 (0.0054 e^{80.3 x_2} + 1)^3 e^{-25 x_2}}{(x_1 + 0.0027)^3}$ |
| 22 | $\dfrac{0.18 x_1 (0.02 e^{80.3 x_2} + 1)^3 e^{-25 x_2}}{(x_1 + 0.0026)^3}$ |
| 23 | $\dfrac{\dfrac{0.19 x_1^2 e^{-25 x_2}}{(x_1 + 0.0027)^3} + 0.13}{x_1}$ |
| 26 | $\dfrac{\dfrac{0.19 x_1^2 e^{-25 x_2}}{(x_1 + 0.0027)^3} + 0.11}{x_1}$ |
| 27 | $\dfrac{x_1 \left( \dfrac{0.19 x_1 e^{-25 x_2}}{(x_1 + 0.0027)^3} + x_2 - 0.91 \right) + 0.13}{x_1}$ |
| 28 | $\dfrac{x_1^2 \left( 0.98 + \dfrac{0.19 e^{-25 x_2}}{(x_1 + 0.0027)^3} \right) - x_1 + 0.14}{x_1}$ |
| 30 | $\dfrac{x_1 \left( x_1 + \dfrac{0.19 x_1 e^{-25 x_2}}{(x_1 + 0.0027)^3} - 1 \right) + 0.14}{x_1}$ |

*Table 22.* Hyper-parameters for symbolic regression on the dijet dataset (using loss SR MSE)

| Hyper-parameter | Value | Notes |
|---|---|---|
| **Neural Spline Flow** | | |
| K | 16 | Number of autoregressive rational quadratic spline flow blocks. |
| hidden units | 128 | Width of the neural networks inside each autoregressive spline flow. |
| hidden layers | 2 | Depth of the neural networks inside each autoregressive spline flow |
| number of samples | 90000 | |
| epochs | 1390 | Number of training epochs |
| batch size | 5000 | batch size of samples used to train flow |
| learning rate | 3e-5 | |
| weight decay | 1e-5 | weight decay regularization |
| **PySR** | | |
| binary operators | [+, -, *, /] | |
| unary operators | [exp, log, pow2, pow3] | Allowed unary operators. |
| maxsize | 30 | Maximum size of an expression tree. |
| ncycles per iteration | 380 | Cycles per iteration of the genetic algorithm. |
| parsimony | 0.001 | Penalty for complexity of expressions. |
| adaptive parsimony scaling | 1040 | Scales the penalty based on frequency of each complexity level |
| niterations for SR | 8000 | Iterations for SR. |
| num populations for SR | 15 | Number of populations for SR. |
| population size for SR | 30 | Number of expressions per population in SR |
| elementwise loss | MSE | Loss function used for SR. |
| batch size | 1000 | batch size of the dataset used to evaluate loss of expressions |

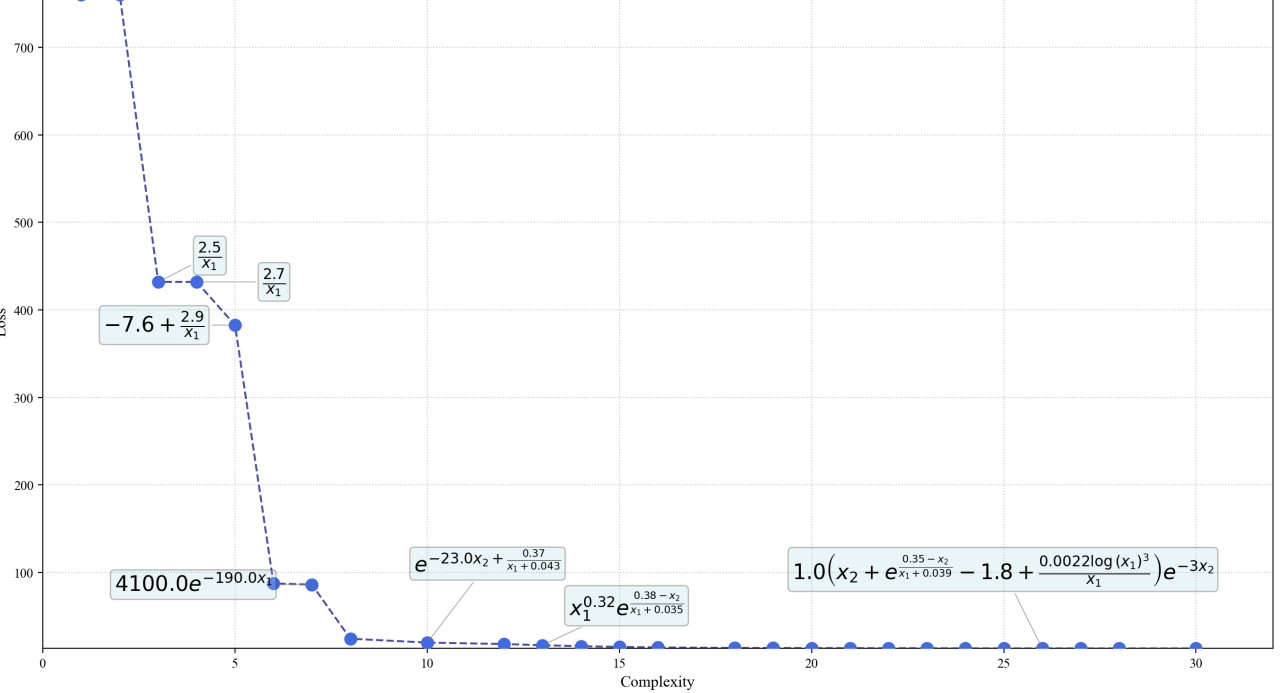

*Figure 22.* Pareto front of recovered expressions for the dijet dataset (SR MSE).

*Table 23.* Expressions simplified with *sympy* alongside their raw complexity score from PySR for the dijet dataset (SR MSE).

| Raw Complexity | Simplified Expression |
|---|---|
| 1 | $0.82$ |
| 2 | $0.82$ |
| 3 | $\frac{2.5}{x_1}$ |
| 4 | $\frac{2.7}{x_1}$ |
| 5 | $-7.6 + \frac{2.9}{x_1}$ |
| 6 | $\frac{0.0027}{(x_1+0.0078)^3}$ |
| 7 | $4100e^{-190x_1}$ |
| 8 | $e^{\frac{0.34-x_2}{x_1+0.04}}$ |
| 10 | $e^{\frac{-23x_2(x_1+0.043)+0.37}{x_1+0.043}}$ |
| 12 | $\frac{x_1e^{\frac{0.35-x_2}{x_1+0.04}}-0.42}{x_1}$ |
| 13 | $x_1^{00.32}e^{\frac{0.38-x_2}{x_1+0.035}}$ |
| 14 | $\frac{x_1e^{\frac{0.35-x_2}{x_1+0.04}}+x_2-0.74}{x_1}$ |
| 15 | $\frac{x_1e^{\frac{0.35-x_2}{x_1+0.039}}+0.14\log(x_1)}{x_1}$ |
| 16 | $\frac{x_1e^{\frac{0.32-x_2}{x_1+0.037}}+0.0022\log(x_1)^3}{x_1}$ |
| 18 | $\frac{x_1\left(e^{\frac{0.32-x_2}{x_1+0.037}}-0.8\right)+0.0022\log(x_1)^3}{x_1}$ |
| 19 | $\frac{x_1\left(e^{\frac{0.32-x_2}{x_1+0.037}}-0.79\right)+0.0022\log(x_1)^3}{x_1}$ |
| 20 | $\frac{x_1\left(x_2+e^{\frac{0.32-x_2}{x_1+0.037}}-1.3\right)+0.0022\log(x_1)^3}{x_1}$ |
| 21 | $\frac{x_1\left(-1.7(1-0.77x_2)^2+e^{\frac{0.32-x_2}{x_1+0.037}}\right)+0.0022\log(x_1)^3}{x_1}$ |
| 22 | $\frac{\left(-x_1\left(1.4-e^{\frac{0.35-x_2}{x_1+0.039}}\right)-0.017\log(x_1)^2\right)e^{-3x_2}}{x_1}$ |
| 23 | $\frac{\left(-x_1\left(1.3-e^{\frac{0.35-x_2}{x_1+0.039}}\right)-0.017\log(x_1)^2\right)e^{-3x_2}}{x_1}$ |
| 24 | $\frac{1\left(x_1\left(e^{\frac{0.35-x_2}{x_1+0.039}}-1.6\right)+0.0022\log(x_1)^3\right)e^{-3x_2}}{x_1}$ |
| 25 | $\frac{0.96\left(x_1\left(e^{\frac{0.35-x_2}{x_1+0.039}}-1.6\right)+0.0022\log(x_1)^3\right)e^{-3x_2}}{x_1}$ |
| 26 | $\frac{1\left(x_1\left(x_2+e^{\frac{0.35-x_2}{x_1+0.039}}-1.8\right)+0.0022\log(x_1)^3\right)e^{-3x_2}}{x_1}$ |
| 27 | $\frac{0.96\left(x_1\left(e^{\frac{0.35-x_2}{x_1+0.039}}-1.6\right)+0.0022\log(x_1)^3\right)e^{-3x_2^2-3x_2}}{x_1}$ |
| 28 | $\frac{0.96\left(x_1\left(e^{\frac{0.35-x_2}{x_1+0.039}}-1.6\right)+0.0022\log(x_1)^3\right)e^{-3x_2^2-3x_2}}{x_1}$ |
| 30 | $\frac{x_1\left(e^{\frac{0.35-x_2}{x_1+0.039}}-1.6\right)+0.0022\log(x_1)^3}{x_1\left(e^{x_2+(x_1+x_2)^2}+0.012\right)^3}$ |

*Table 24.* Hyper-parameters for symbolic regression on the dijet dataset (using loss SR NLL + NP)

| Hyper-parameter | Value | Notes |
|---|---|---|
| **Neural Spline Flow** | | |
| K | 16 | Number of autoregressive rational quadratic spline flow blocks. |
| hidden units | 128 | Width of the neural networks inside each autoregressive spline flow. |
| hidden layers | 2 | Depth of the neural networks inside each autoregressive spline flow |
| number of samples | 90000 | |
| epochs | 1390 | Number of training epochs |
| batch size | 5000 | batch size of samples used to train flow |
| learning rate | 3e-5 | |
| weight decay | 1e-5 | weight decay regularization |
| **PySR** | | |
| binary operators | [+, -, *, /] | |
| unary operators | [exp, log, pow2, pow3] | Allowed unary operators. |
| maxsize | 30 | Maximum size of an expression tree. |
| ncycles per iteration | 380 | Cycles per iteration of the genetic algorithm. |
| parsimony | 0.001 | Penalty for complexity of expressions. |
| adaptive parsimony scaling | 1040 | Scales the penalty based on frequency of each complexity level |
| niterations for SR | 8000 | Iterations for SR. |
| num populations for SR | 15 | Number of populations for SR. |
| population size for SR | 30 | Number of expressions per population in SR |
| elementwise loss | NLL + NP | Loss function used for SR. |
| batch size | 1000 | batch size of the dataset used to evaluate loss of expressions |

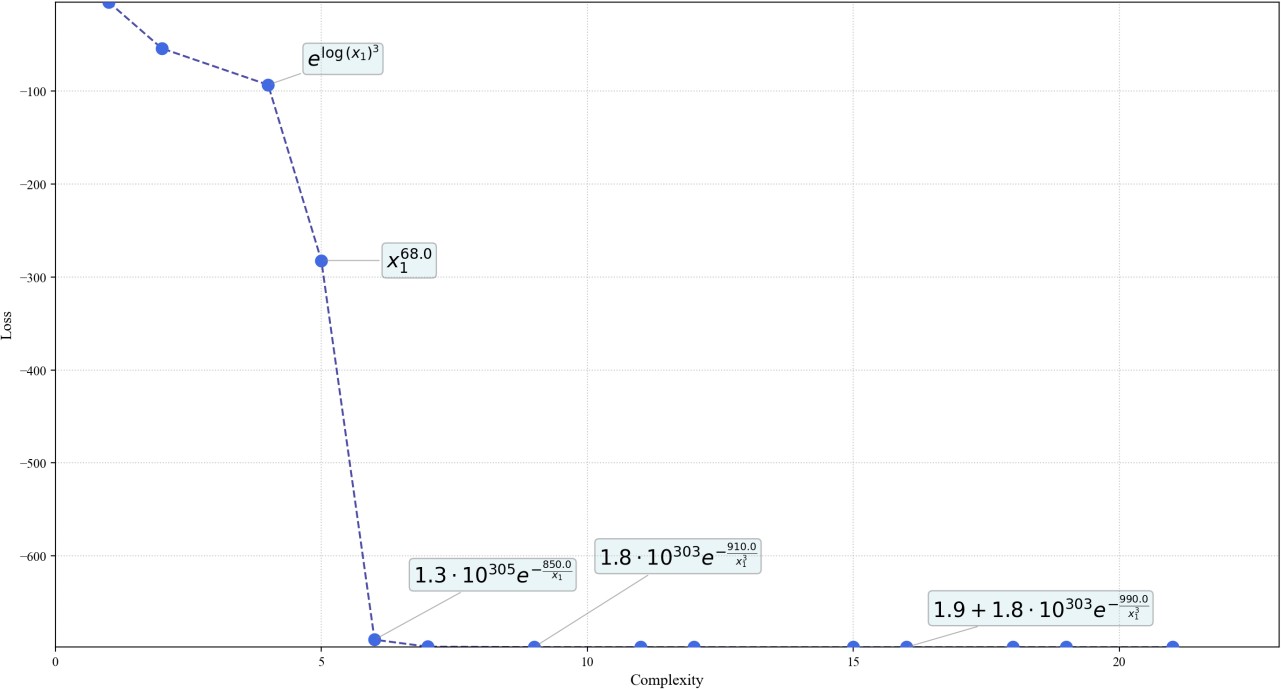

*Figure 23.* Pareto front of recovered expressions for the dijet dataset (SR NLL + NP).

*Table 25.* Expressions simplified with *sympy* alongside their raw complexity score from PySR for the dijet dataset (SR NLL + NP).

| Raw Complexity | Simplified Expression |
|:---:|:---|
| 1 | $x_1$ |
| 2 | $e^{x_2}$ |
| 4 | $e^{(\log(x_1))^3}$ |
| 5 | $x_1^{68}$ |
| 6 | $1.3 \cdot 10^{35} e^{-\frac{850}{x_1}}$ |
| 7 | $1.3 \cdot 10^{33} e^{-\frac{910}{x_1^3}}$ |
| 9 | $1.8 \cdot 10^{33} e^{-\frac{910}{x_1^3}}$ |
| 11 | $0.4 + 1.8 \cdot 10^{33} e^{-\frac{910}{x_1^3}}$ |
| 12 | $1.9 + 1.8 \cdot 10^{33} e^{-\frac{1000}{x_1^3}}$ |
| 15 | $1.9 + 1.8 \cdot 10^{33} e^{-\frac{1000}{x_1^3}}$ |
| 16 | $1.9 + 1.8 \cdot 10^{33} e^{-\frac{990}{x_1^3}}$ |
| 18 | $1.9 + 1.8 \cdot 10^{33} e^{-\frac{990}{x_1^3}}$ |
| 19 | $1.9 + 1.8 \cdot 10^{33} e^{-\frac{1000}{x_1^3}}$ |
| 21 | $1.9 + 1.8 \cdot 10^{33} e^{-\frac{990}{x_1^3}}$ |

