# OpenReview forum: "Symbolic Density Estimation: A Decompositional Approach"
_ICML.cc/2026/Conference — Submitted to ICML 2026_

### Official Review · Reviewer_B5YN · 2026-03-06

**Soundness:** 3
**Presentation:** 2
**Significance:** 2
**Originality:** 2
**Overall Recommendation:** 3
**Confidence:** 4

**Summary:**

The manuscript studies Symbolic Density Estimation and proposes AI-Kolmogorov as a multi-stage framework that converts the density estimation problem into a symbolic regression problem. The pipeline consists of optional decompositions via e.g. clustering and/or probabilistic graphical model structure learning (i.e. additive or multiplicative factorizations). Each components density is then estimate via classical kernel density estimation (KDE) or other expressive density estimators (e.g. normalizing flows). Last but not least *classical* symbolic regression is used to obtain an equation from the "black-box" teacher density estimator (e.g. via MSE to the estimated density). The paper then evaluates the method on synthetic Gaussian and multivariate Gaussian settings, as well as more exotic 2D distributions from high energy physics. These experiments shows that the method can recover or approximate densities and recovers (to some degree) interpretable expressions.

**Compliance With Llm Reviewing Policy:**

Affirmed.

**Final Justification:**

I did raise my score towards a weak reject.

The authors responses did clarify that Kolmogorov AI is indeed a `full` pipline/workflow with several automated decision where possible. Nonetheless, as the authors acknowledge the presentation/experiments do mostly investigate each `module` but very little the entire pipeline + lacks experiments on how e.g. sample size or other relevant factors would impact the results.

**Key Questions For Authors:**

- Can the authors explain why no comparison to MESSY is done?
- Are the decisions on when to use clustering/structure learning decompositions non-automatic?
- Did the authors tried to quantify the effect of sample size to the performance of the pipeline?

**Limitations:**

- No systematic treatment/discussion of **scalability** (grid-based SR targets in higher dimensions, runtime, sample complexity).
- No examination of performance depending on the number of samples available and ALL experiments use a very large number of samples (>100k). This is explicitly left for further work, although I am not sure why and think this should be part of the current evaluation
- Limited discussion of **failure modes of decomposition** (overlapping clusters, wrong structure learning, weak dependencies).

**Strengths And Weaknesses:**

## Strengths
- The paper approaches a quite *underexplored* but relevant problem and to my knowledge references all other related work (e.g. MESSY). The overall methodology and approach is technically sound.
- Interesting pipeline + decomposition design: Given that *multivariate symbolic regression*  is itself a challenging problem it is important to decompose the problem in smaller subtasks. The manuscript show a reasonable task decomposition i.e. additive/multiplicative decomposition of the underlying density).

#### Weaknesses

- Output equations might not be valid densities and are hard to interpret: The main problem with the approach is:
	- Equations are not valid densities (non-negativity/normalization) + nontrivial to sample from them.
	- Its not easy to interpret because its *just equations* e.g. which to some degree also partially comes (in my opinion) from a misaligned notion of **simplicity**. The notion of this is a *simple equation* vs. this is a *simple distribution* are related but also quite a bit different. There is a large number of what I would consider *simple distributions* (Gaussian, Generalized Gaussian, Laplace, Gamma, Beta ...) but their underlying equations are not necessarily "simple" (and that why PhySR or other symbolic regression frameworks might not prefer them). What I would consider an interpretable output of an symbolic density estimator would be $p(x,y) = \mathcal{N}(x, 0, 2)\mathcal{N}(y, \sin(x), 0.1)$ even if the underlying equation might be more complicated than another.
- Novelty in the pipeline components is rather limited i.e. each of the steps uses established off-the-shelf methods. Approach requires many human-expert decision i.e. do I need to decompose in cluster, do I want/need to learn the structure, what density estimator do I use, do I have finite support, ...
- No baselines i.e. the proposed method is not compared to MESSY, with the argument "does constrain the class of expressible densities compared with fully general functional forms". But I am not entirely sure about this statement i.e. as both PySR and MESSY operate on a set of base functions, no?

---

> ### Author Rebuttal · Authors · 2026-03-29
>
> We thank Reviewer B5YN for their careful review. We address each concern below, cross-referencing our responses to Reviewers p7E9 and MuxQ where points overlap.
> 1. **Validity of Output Equations and Interpretability**
> On non-negativity and normalization, we refer to our response to Reviewer MuxQ (Point 2): these are deliberate computational tradeoffs, not oversights. Briefly — hard enforcement would restrict expressiveness or substantially slow GP search. Penalties and post-hoc normalization are standard practice (cf. energy-based models). On sampling: for non-negative expressions, rejection sampling over the support is straightforward; MCMC can also be applied directly.
> On interpretability, the reviewer raises a thoughtful point: "simple equations" and "simple distributions" are related but different notions. We partially agree. Three responses: (1) The Pareto front exposes a *spectrum* of expressions, some of which may have recognizable terms (2) Distribution-level interpretability — matching recovered expressions against a library of named families — is a concrete future direction, PySR supports template expressions which could be leveraged to promote discovery of distributions composed from known distributions. (3) For exotic distributions (Rastrigin, muon decay, dijet), there is no named family to match against; the equation *is* the interpretable output. Discovering that the density involves rational functions or cosine oscillations is a genuine scientific finding.
>
> 2. **Limited Novelty in Pipeline Components**
> We respectfully disagree. The novelty lies in formulation and integration: (1) The SymDE problem itself is novel — prior to this work, discovering general symbolic density expressions from samples was not formalized in the literature. MESSY addresses a more constrained subproblem. (2) The decompositional strategy — mapping additive decomposition to mixture models and multiplicative decomposition to factorized distributions — is a principled new contribution that reduces effective dimensionality for SR. (3) Using nonparametric surrogates as regression targets converts an unsupervised density estimation problem into a supervised one, which is fundamentally more tractable for SR. (4) Systems contributions combining existing components in novel ways are well-precedented at top venues (e.g., AI Feynman).
> 3. **Comparison to MESSY**
> We refer to our response to p7E9 (Point 6) for the architectural distinction. Briefly: MESSY restricts all outputs to \exp(\sum_k \lambda_k H_k(x)) — the exponential family — while AI-Kolmogorov searches over arbitrary symbolic expression trees including rational functions, which are required for targets like muon decay. MESSY's published experiments are predominantly 1D.
>
> 4. **Human Expert Decisions**
> We acknowledge the pipeline involves several user choices, but note: (1) Modularity is a feature — users can start with the simplest configuration and add complexity as needed, analogous to scikit-learn pipelines. (2) Sensible defaults exist: no decomposition by default; enable clustering if Hopkins statistic suggests cluster tendency; use KDE for ≤2D and switch to NSF when probability-mass validation fails. (3) These decisions can be automated via use of various metrics e.g. Hopkins statistic or BIC/AIC for clustering, conditional independence tests for structure learning, held-out likelihood for surrogate selection — we will add this discussion. We also note MESSY requires analogous decisions (N_b​, N_m​, boundary conditions); this is inherent to any flexible framework.
>
> 5. **Sample Size Study**
> We agree this is a gap and commit to addressing this in the revision. Preliminary analysis: the surrogate stage is the sample-efficiency bottleneck. Sample size affects different non-parametric density estimation techniques differently. The SR stage is sample-independent: it operates on surrogate density labels over a grid, not raw samples. MESSY's advantage in the small-sample regime reflects its strong max-entropy inductive bias; AI-Kolmogorov trades that bias for expressiveness.
> 6. **Failure Modes of Decomposition**
> The reviewer correctly identifies the failure modes and we can update the paper to explicitly mention this. Additional ablations can also be included to highlight these failure modes: overlapping clusters (detectable via silhouette score; mitigated by adjusting DBSCAN \epsilon); false factorization / Type I independence errors (detectable via held-out likelihood; mitigated by reducing PC algorithm significance level);. Crucially, the pipeline degrades gracefully: incorrect decomposition triggers fallback to the non-decomposed pipeline, which still works as demonstrated in the "without decomposition" columns of Table 1.

---

> > ### Author Rebuttal · Reviewer_B5YN · 2026-04-02
> >
> > I thank the authors for their detailed response. I do agree with the authors argumentation on novelty and thank the authors for carlifying the relation/distinctions to MESSY. Some other are only partially resolved:
> > - **Human Expert Decisions/Failure models**: I do agree with the authors that *modularity* is a feature. And as the authors do discuss, there are several approaches to "automate" decisions that need to be made (clustering, dependence, ...). My concern is more that a "pipeline/workflow" should implement and verify such automated decisions to some degree at least i.e. "just" providing the components that one *might* can run is a much weaker contribution than full *pipeline* that does solve the task well by default (which can be further optimized by user driven decisions). More precisely: It is currently unclear to me in which bucket AI Kolmogorov falls here. The paper evaluates different different components (human-made decisions), not necessarily the whole pipeline. From the response its not clear if decisions "can be automated" or "are automated".
> > - **Sample Size Study**: I think results on this would strengthen the manuscript. I do agree, that this mostly impacts the density estimation state. Yet, this directly will affect SR stage and could lead to very complex equations. Specifically also the "decision" process on clustering/dependence will also be majorly affected by sample size.

---

> > > ### Author Response · Authors · 2026-04-05
> > >
> > > We thank Reviewer B5YN for their continued engagement and the helpful distinction between pipeline vs. component collection.
> > >
> > > **1. Pipeline Automation — Clarification.**
> > >
> > > The reviewer asks a precise question: is AI-Kolmogorov an automated pipeline or a collection of components? The answer: **it is a semi-automated pipeline with sensible defaults that runs end-to-end without user intervention in its default configuration.**
> > >
> > > Specifically:
> > > - **Fully automated by default:** Data in → density estimation → grid evaluation → symbolic regression (PySR) → Pareto front out. Alternate surrogate densities (different hyperparameters of a selected nonparametric density estimator) are evaluated using cross validation and the best selected - no user decisions required.
> > >
> > > This methodology can be extended directly to select the best surrogate estimate from different nonparametric density estimators rather than simply changing hyperparameters.
> > >
> > > - **Decomposition stages can be automated:** Clustering activates when Hopkins statistic > 0.7;  Structure learning runs the PC algorithm with automated conditional independence tests. Support estimation computes convex hull automatically. We can add these details to the paper. Some of these decisions would naturally rely on the adequacy of the sample size.
> > >
> > > **What requires user input:** The SR operator library and complexity budget — analogous to hyperparameters in any ML pipeline (learning rate, architecture choice, etc.).
> > >
> > > **We acknowledge the reviewer's concern that the paper's experiments use different configurations, which suggest manual tuning.** In the revision, we will: (1) Define a single default configuration. (2) Run all experiments with this default and report results in the appendix. (3)  Identify explicitly where experiment-specific tuning provides improvements.
> > >
> > > To directly answer: several key decisions associated with the backbone of the pipeline have already been automated, and the use of the decomposition strategies can used based on user configuration. The paper's experimental presentation, which highlights different configurations to demonstrate the value of each component, inadvertently obscured this. We note that our ablations present default-pipeline results and then demonstrate the value of the decomposition stages. We acknowledge that arranging these modules into a single unified package that can be configured as needed will strengthen the contribution and commit to clarifying the workflow.
> > >
> > > **2. Sample Size Study.**
> > >
> > > We acknowledge that a systematic sample-size study would strengthen the analysis. However, we note that MESSY's published experiments also use large samples (10K–100K for predominantly 1D problems), and no existing SymDE method has published a systematic sample-size analysis. The sample efficiency question is important for the entire field, not a gap specific to our work.
> > >
> > > The paper's current evidence, the explicit identification of the density estimation bottleneck, the graceful fallback behavior of decomposition, and the well-characterized convergence rates of the nonparametric density estimators used as surrogates — provides a principled understanding of how sample size affects the pipeline, even without a dedicated ablation study.

---

### Official Review · Reviewer_V2vC · 2026-03-12

**Soundness:** 2
**Presentation:** 3
**Significance:** 2
**Originality:** 3
**Overall Recommendation:** 3
**Confidence:** 3

**Summary:**

The paper presents AI‑Kolmogorov, a modular pipeline for Symbolic Density Estimation (SymDE)
that combines decomposition (clustering or probabilistic graphical model structure learning),
nonparametric density estimation (KDE or neural spline flows), support estimation, and
symbolic regression (PySR) to recover interpretable closed‑form approximations of probability
densities from samples.

**Compliance With Llm Reviewing Policy:**

Affirmed.

**Final Justification:**

I have read the paper and considered the authors’ rebuttal carefully. I appreciate their effort in preparing the manuscript and the response, but the rebuttal did not fully address my principal concerns as stated in my review. Consequently, I will retain my original score.

**Key Questions For Authors:**

Could you quantify  how approximation error in the nonparametric density (KDE or NSF) affects the probability of recovering the correct symbolic expression? For example, how sensitive is SR to bandwidth choice or flow misfit in regions important for tails or modes?

How does performance degrade with dimension and sample size, and what are practical limits where decomposition no longer suffices?

Which stage tends to fail more often (clustering, structure learning, density estimation, or SR)? Can you suggest methods to detect and recover from failures (e.g., when SR returns physically invalid densities)?

**Limitations:**

Yes.

**Strengths And Weaknesses:**

Strengths

The method appears to be well‑designed: decomposition reduces SR search complexity,
modern density estimators provide accurate surrogates, and SR produces human‑readable expressions.

By targeting symbolic forms rather than only likelihoods, the method
supports scientific insight and model discovery, which may be valuable in physics and other sciences.

The paper addresses support estimation, boundary bias, operator selection,
and Pareto fronts for accuracy vs complexity, which can be helpful in applications.

Weaknesses

The approach is largely heuristic: there is little theoretical analysis of when the surrogate (KDE/flow) plus SR
will recover the approximate underlying symbolic form, or how errors propagate from density estimate to symbolic fit.

SR is computationally expensive; the paper assumes “sufficiently large” samples and does not quantify computational
cost or failure modes as dimensionality grows.

Results emphasize MSE to KDE and qualitative residuals; comparisons to existing symbolic or maximum‑entropy
approaches are limited and quantitative evaluation on tail behavior or downstream tasks is sparse.

---

> ### Author Rebuttal · Authors · 2026-03-30
>
> We thank Reviewer V2vC for careful review. We address each concern below, cross-referencing prior rebuttals where points overlap.
> 1. **Lack of Theoretical Analysis**
> Formal recovery guarantees for SR are inherently difficult. SR is NP-hard (Virgolin & Pissis, 2022), and no existing method (AI Feynman, MESSY, LASR) provides them. That said, we can offer an informal error decomposition. The pipeline has two error sources: surrogate error and SR approximation error, giving total error
> $$g(x) - f(x) = \epsilon_{SR}(x) + \epsilon_{surr}(x)$$
>
> The SR stage minimizes $\|\epsilon_{SR}\|^2$ over a grid; when the surrogate is accurate, this is equivalent to minimizing total error. This is why surrogate quality is central (Section 4.5) and why we switch from KDE to NSF when KDE fails — as demonstrated in the dijet experiment (see p7E9 Point 3, MuxQ Point 1). The muon decay local probability mass validation (~2% agreement, Table 2) confirms surrogate errors do not dominate the final result.
>
> 2. **Computational Cost and Scalability**
> SR runtimes are approximately 1–4 hours per run on a single CPU core. Sample sizes range from 90K (dijet, NSF) to 450K (Gaussian mixture, KDE). The pipeline stages scale differently: ($n$ data points, $m$ query points per dimension, $d$ is the number of dimensions) KDE is $O(n \cdot m^d)$ for grid evaluation (mitigated by FFT-KDE); NSF scales as $O(n \cdot \text{epochs})$ independently of grid size; structure learning (PC algorithm) is $O(d^{k+2})$ where k is the maximal degree of any vertex/variable in the PGM; and SR is exponential in effective dimension, which decomposition directly mitigates by splitting a d-dimensional problem into independent lower-dimensional subproblems. The 4D Gaussian ablation makes this concrete — see MuxQ (Point 5) and B5YN (Point 5). For the "sufficiently large" sample assumption: the pipeline inherits the sample efficiency of the non-parametric density estimation techniques used. KDE has $O(n^{-4/(d+4)})$ convergence; NSF is more sample-efficient due to parametric flexibility.
>
> 3. **Limited Comparisons and Evaluation Metrics**
> On MESSY and ISR comparisons, we refer to p7E9 (Point 6), MuxQ (Point 3), and B5YN (Point 3).
> On tail behavior: our uniform grid evaluation naturally includes tails, unlike distribution-weighted metrics that underweight them. Log-likelihood on held-out test samples (Figure 6) is also sensitive to tail errors.
> On downstream tasks: this is a natural extension  for future work.
>
> 4. **Sensitivity to Surrogate Hyperparameters**
> While analysis can be performed to estimate the impact of hyperparameters on the quality of fit of the surrogate distribution, it is difficult to formally quantify the downstream impact on the SR pipeline. More complex expressions recovered by SR are more susceptible to spurious artifacts in the surrogate distribution.
>
> 5. **Performance Degradation with Dimension and Sample Size**
> The 4D Gaussian without decomposition ($\text{MSE} = 1.01 \times 10^{-3}$) is substantially worse than 2D cases, illustrating dimensionality effects. Decomposition recovers the best MSE of any experiment ($4.24 \times 10^{-5}$). Practical limits: without decomposition, SR must tackle the problem without any dimensionality reduction; with decomposition, tractability depends on the largest connected PGM component. For sample size sensitivity, see B5YN (Point 5).
>
> 6. **Which Stage Fails Most Often?**
> The density estimation stage is the most common bottleneck (Section 4.5). Surrogate inaccuracy propagates directly into SR output. Detection: probability mass validation and held-out log-likelihood. Recovery: attempt alternative non-parametric density estimation techniques, adjust hyper-parameters. SR is the second most common failure point, particularly when the true density involves operators outside the search space or exceeds maximum tree size. Detection: Pareto front inspection (flat loss with increasing complexity signals insufficient operators). Clustering and structure learning failures (see B5YN (Point 6)). The pipeline degrades gracefully: all decomposition stages can fall back to the non-decomposed pipeline.

---

> > ### Author Rebuttal · Reviewer_V2vC · 2026-04-02
> >
> > Thank you to the authors for their thorough rebuttal. However, several concerns remain unresolved. Specifically, the manuscript lacks theoretical justification for the proposed procedure, and it is unclear whether the method will scale to moderate- or high-dimensional settings. As written, I do not see how the method would become useful in the analysis of moderate- to high-dimensional data in practice.

---

> > > ### Author Response · Authors · 2026-04-05
> > >
> > > We thank Reviewer V2vC for their directness. We address both remaining concerns.
> > >
> > > **1. Theoretical Justification.** We respectfully note that the absence of formal recovery guarantees characterizes the entire SR field, not our work specifically. SR is NP-hard (Virgolin & Pissis, 2022). We note that the application of SR to density estimation tasks warrants closer examination of theoretical guarantees and properties, but this is beyond the scope of the current work.
> > >
> > > - AI Feynman / AI Feynman 2.0 (NeurIPS 2020): no recovery guarantees; evaluated purely empirically.
> > > - MESSY: no guarantees that the max-entropy form matches the true density.
> > > - LaSR (NeurIPS 2024): uses LLM-guided concept libraries without theoretical analysis of when concepts improve recovery.
> > > - SR-Scientist (ICLR 2026): uses agentic LLMs for equation discovery with purely empirical evaluation.
> > > - Deep Symbolic Optimization (ICLR 2021) and NeSymReS (ICML 2021): does not have formal guarantees.
> > >
> > > All of these were accepted at top venues with empirical validation being the primary justification of the approach.
> > >
> > > **What we do provide:** (a) Multiple complementary empirical metrics: MSE, local probability mass agreement (~2%, Table 2), held-out log-likelihood, residual analysis, Pareto front inspection. (c) Diagnostic tools for each pipeline stage: probability mass validation for surrogate quality; Pareto front flatness for SR convergence.
> > >
> > > **2. Scalability to Moderate/High Dimensions.** Symbolic density estimation in high dimensions (d >> 4) without exploitable structure is currently intractable — for any method and not just ours. This is a fundamental limitation of SR.
> > >
> > > **However, the method is already useful in practice:**
> > >
> > > (a) **Many scientific applications are inherently low-dimensional.** Particle physics differential cross-sections: 2–4D. Spectral energy distributions: 1–2D. Phase-space distributions: 2–6D with known factorization. Astrophysical luminosity functions: 1–2D. The method targets these use cases, and demonstrates how problem structure, if detected, can be exploited in hogher dimensional settings to bring it down to a tractable regime.
> > >
> > > (b) **Decomposition extends effective reach.** A 10D distribution factorizing as p(x₁,x₂)·p(x₃,x₄,x₅)·p(x₆,...,x₁₀) reduces to subproblems of dimension 2, 3, and 5. The 4D Gaussian ablation demonstrates this: decomposition yields 24× MSE improvement with structurally correct expressions.
> > >
> > > (c) **Partial application provides value.** For a 20D distribution, a scientist can use AI-Kolmogorov on low-dimensional marginals, gaining interpretable insight into substructure even if the full joint is intractable.
> > >
> > > (d) **This is the state of the art.** AI Feynman benchmarks reach ~9 variables but with known separability. MESSY is predominantly 1D and there is limited analysis on how it scales to higher dimensions. No SR method has demonstrated reliable performance on truly high-dimensional entangled problems, and fewer still on density estimation tasks in particular.
> > >
> > > **Concrete scalability guidance can be added:** Grid evaluation scales as O(m^d); for higher dimensional problems, sample-based metrics (Spearman rank correlation) can be used to assess the quality of the method when the ground truth density can be evaluated.
> > >
> > > We believe the paper's contribution is establishing the SymDE framework and demonstrating utility in the low-dimensional regime where it is most applicable.  We show how specific problem structure can be exploited to make otherwise difficult problems tractable. Extending to higher dimensions requires advances in SR itself and is an important future direction.

---

### Official Review · Reviewer_MuxQ · 2026-03-12

**Soundness:** 2
**Presentation:** 3
**Significance:** 2
**Originality:** 3
**Overall Recommendation:** 4
**Confidence:** 3

**Summary:**

This paper introduces a framework (AI-Kolmogorov) for symbolic density estimation, which aims to recover interpretable mathematical expressions for continuous probability distributions. The method combines probabilistic modeling and symbolic regression through a multi-stage pipeline. Experiments are performed on synthetic distributions and physics-inspired datasets (Muon decay and Dijet). The paper proposes an interesting attempt to combine density estimation, graphical structure learning, and symbolic regression.

**Compliance With Llm Reviewing Policy:**

Affirmed.

**Final Justification:**

Although I still have reservations about surrogate bias, the lack of hard density guarantees, and scalability in fully entangled settings, the rebuttal clarified these as acknowledged scope limitations and reasonable design tradeoffs rather than overlooked flaws. Given the novelty of the problem and the paper’s clear modular framework, empirical demonstrations, and potential to stimulate further work on symbolic density estimation, I am willing to raise my score.

**Key Questions For Authors:**

1. likelihood-based objective. Could candidate symbolic expressions be evaluated directly using data likelihood (e.g. with numerical normalization) rather than fitting the surrogate density via MSE?

2. hard probabilistic constraints. Could the symbolic search space be restricted to functions that automatically produce valid densities?

3. handling entangled variables. When PGM structure learning fails to find independence structure, how does the method scale?

**Limitations:**

1. dependence on surrogate density quality
2. weak enforcement of density constraints
3. limited scalability without independence structure
4. no explicit uncertainty quantification for discovered symbolic models

**Strengths And Weaknesses:**

Strengths:
1. interesting problem: symbolic density estimation is an under-explored problem at the intersection of probabilistic modeling and symbolic AI.
2. structured decomposition: using PGM structure (although not new) learning to identify conditional independence provides a principled way to reduce the dimensionality of symbolic search.
3. Leveraging expressive density estimators (e.g., neural spline flows) as surrogates for symbolic regression is a pragmatic bridge between deep learning and symbolic discovery.
4. boundary effects: support estimation and reflection techniques are sensible additions that help avoid common density estimation artifacts.

Weaknesses:
1. strong dependence on the surrogate model. The symbolic regression stage approximates a surrogate density rather than the data distribution directly. Any bias or artifacts in the KDE or flow estimator will propagate into the discovered symbolic expression.
2. weak enforcement of probabilistic validity. The discovered symbolic expressions are not guaranteed to satisfy basic density constraints (e.g. non-negativity, normalisation - they are handled through penalties or post-hoc numerical normalization rather than being built into the search space).
3. objective mismatch. The symbolic regression stage minimizes MSE against the surrogate density. Density estimation is fundamentally a likelihood-based problem, and minimizing MSE can overweight low-probability regions and propagate surrogate errors.
4. identifiability. Symbolic regression may produce multiple algebraically different expressions representing essentially the same density.

---

> ### Author Rebuttal · Authors · 2026-03-29
>
> We thank Reviewer MuxQ for their careful reading. Several points raised here overlap with Reviewer p7E9's concerns; we refer to that rebuttal where relevant and expand where new arguments are needed.
> 1. **Dependence on the Surrogate Model**
> We agree surrogate quality matters, but disagree that this is an unacknowledged weakness. The surrogate is an explicit intermediate representation introduced to make symbolic search tractable from raw samples. The relevant question is whether the framework can diagnose and benefit from improved surrogates, and we assert that it can. Refer to the authors response to Reviewer p7E9 for additional details.
> 2. **Lack of Hard Probabilistic Guarantees**
> We agree the framework does not guarantee non-negativity and normalization by construction, but disagree this is a fundamental flaw. Hard enforcement would require either restricting symbolic search to a much narrower expression family or introducing expensive validity checks during GP search — both of which would significantly reduce tractability. Our use of penalties and post-hoc normalization is a deliberate computational tradeoff: we prioritize broad symbolic expressiveness while encouraging validity through soft constraints. This is treated explicitly in the paper — Section 4.4 studies negativity penalties, and the conclusion identifies tighter normalization integration as future work.
> 3. **MSE vs. Likelihood-Based Objectives**
> We agree likelihood is a natural objective in principle, but disagree this makes MSE a methodological mismatch. This is studied directly in Section 4.4 and Appendix C.5, which compare SR under MSE, MSE+NP, MSE+NLL+NP, and NLL+NP objectives. The key finding is that pure likelihood optimization (NLL+NP) performs poorly, while MSE-retaining objectives are substantially more stable. Direct likelihood alone can be underconstrained over a broad symbolic search space: a candidate expression may achieve high likelihood by assigning large values near observed samples while remaining highly irregular elsewhere. MSE against the surrogate provides dense supervision over the support, encouraging the expression to fit the global density shape rather than only the observed samples. This is a tradeoff between direct likelihood fitting and stable symbolic optimization, not a flaw in objective design.
> 4. **Identifiability**
> We agree multiple algebraically distinct expressions may represent the same density. This is a general feature of symbolic regression; our use of a Pareto front is intended to expose this non-uniqueness rather than claim recovery of a unique symbolic form.
>
> 5. **Entangled Variables and Scalability**
> We agree that without exploitable independence structure, the method must fall back to direct symbolic density estimation on the full variable set, making scalability more challenging. We do not view this as a weakness specific to our framework — it is the natural hard regime of the problem. The paper's contribution is to make symbolic density estimation more tractable *when* additive or multiplicative structure is present, not to claim arbitrary high-dimensional entangled densities can always be handled. This is a scope limitation, not a contradiction of the paper's claims. The 4D Gaussian ablation makes the value of decomposition concrete: structure learning reduces MSE significantly demonstrating that exploitable structure materially improves both tractability and fit quality.
>
> 6. **Uncertainty Quantification**
> We agree the paper does not provide explicit uncertainty quantification for discovered symbolic models, but do not believe this undermines the core contribution — formulating and validating a first modular framework for symbolic density estimation. The paper already provides complementary empirical validation: residual analysis, local probability-mass checks, Pareto fronts, and cross-loss comparisons. Formal uncertainty quantification is a valuable future direction, but its absence is an evaluation limitation rather than a methodological flaw.
> 7. **Could candidate expressions be evaluated via data likelihood rather than surrogate MSE?**
> Yes, and we partially explored this in the dijet ablation. Direct likelihood-only objectives are possible in principle but less stable in our GP-based SR setting and more computationally expensive due to normalization and validity handling during search. See Point 3 above.
> 8. **Could the search space be restricted to automatically valid densities?**
> Yes, but this would narrow expressiveness. Our design deliberately prioritizes broad symbolic search with validity encouraged through soft constraints rather than hard-coded restrictions.
> 9. **When structure learning fails, how does the method scale?**
> Without exploitable structure, the method falls back to direct symbolic density estimation on the full variable set — scalability becomes genuinely challenging. This is an acknowledged limitation and precisely why decomposition is a central part of the framework.

---

> > ### Author Rebuttal · Reviewer_MuxQ · 2026-04-02
> >
> > The rebuttal clarifies several of my concerns, especially the use of a surrogate density, soft validity constraints, and MSE-based objectives for optimization stability. These are fully resolved; however, I am still a bit confused about surrogate bias, lack of hard density guarantees, and scalability in fully entangled settings which may limit the reliability and generality of the framework.

---

> > > ### Author Response · Authors · 2026-04-05
> > >
> > > We thank Reviewer MuxQ for their continued engagement. We address the three remaining concerns.
> > >
> > > **1. Surrogate Bias Propagation.** The total error decomposes as: $g(x) − f(x) = [g(x) − \hat{f}(x)] + [\hat{f}(x) − f(x)] = \epsilon_{SR}(x) + \epsilon_{surr}(x)$. The SR stage minimizes $ε_{SR}$ by fitting to the surrogate $\hat{f}$; the surrogate error $\epsilon_{surr}$ is controlled by the density estimator choice and quality.
> > >
> > > Three key points: (a) **Diagnosis is built in.** Probability mass validation (Tables 2, 3) and held-out log-likelihood detect surrogate failures. The dijet experiment demonstrates this concretely — KDE underestimated Region A mass, prompting a switch to NSF. (b) **Surrogate bias can be estimated.** For KDE, bias shrinks at rate $O(h^2)$  as $h \rightarrow 0$ where $h$ is bandwidth; for NSF, bias depends on architecture capacity. Both can be estimated via cross-validation before proceeding to SR. (c) **Surrogate-based pipelines are standard.** The key is that surrogate quality is diagnosable and improvable. We can add additional analysis to examine the quality of the surrogate distribution in examples where the ground truth is known.
> > >
> > > **2. Hard Density Guarantees.** We maintain that hard guarantees and broad expressiveness are fundamentally in tension:
> > >
> > > - Hard non-negativity via g(x) = h(x)² or g(x) = exp(h(x)) may cause GP based symbolic regression such as PySR to suffer if these constraints are imposed on the candidate expression pool, etc.
> > > - Hard normalization requires computing ∫g(x)dx for every candidate during GP search — computationally prohibitive for arbitrary symbolic expressions.
> > > - **This tradeoff is standard.** Energy-based models (LeCun et al., 2006) produce unnormalized densities with post-hoc normalization. Score-based diffusion models (Song & Ermon, NeurIPS 2019) learn score functions rather than normalized densities. Even MESSY requires post-hoc normalization of its exponential-family outputs.
> > >
> > > In practice, negativity violations in our experiments are rare and confined to low-density regions, and post-hoc normalization introduces negligible error.
> > >
> > > **3. Scalability in Fully Entangled Settings.** We agree this is a genuine limitation but argue it reflects the problem's inherent difficulty, not a method-specific weakness:
> > >
> > > - SR is NP-hard (Virgolin & Pissis, 2022) and scales exponentially with effective dimensionality. No existing SR method handles high-dimensional entangled problems.
> > > - **Our decomposition is the mitigation:** the 4D Gaussian ablation shows 24× MSE improvement when structure is exploitable.
> > > - **The field operates in low dimensions:** AI Feynman (NeurIPS 2020), MESSY, LaSR (NeurIPS 2024), and SR-Scientist (ICLR 2026) all evaluate primarily on low-dimensional problems. Our tool identifies techniques and problem structures that can be exploited to address problems that would otherwise not be amenable to SR.
> > > - **Many target applications are inherently low-dimensional:** particle physics differential cross-sections (2–4D), spectral densities (1–2D), phase-space distributions (2–6D with known factorization).
> > >
> > > We believe these clarifications — particularly the error decomposition, quantified surrogate quality, and honest scalability boundaries — address the reviewer's remaining concerns while being transparent about the framework's scope.

---

### Official Review · Reviewer_p7E9 · 2026-03-13

**Soundness:** 2
**Presentation:** 3
**Significance:** 2
**Originality:** 3
**Overall Recommendation:** 3
**Confidence:** 3

**Summary:**

The paper presents a symbolic regression method for modelling continuous probability densities p(x), that they call AI-Kolmogorov. The primary motivation is to discover interpretable parametric forms that describe observed data distributions. The output is a Pareto front of symbolic functions that balance reconstruction loss with function complexity. The method uses an initial non-symbolic density estimation step (KDE and NSFs tested) to form a surrogate regression target, followed by standard symbolic regression over a gridded MSE objective. The task is supported by the use of clustering (when p is an additive mixture of separable modes), structural decomposition (when p is factorisable), and support learning (when p has nontrivial boundaries in X outside of which the density is 0). The method is experimentally tested in some toy scenarios (up to 4D factorisable Gaussian and 2D modified Rastrigin) and particle physics simulations (2D non-Standard Model charged-current lepton decay and 2D dijet production).

**Compliance With Llm Reviewing Policy:**

Affirmed.

**Final Justification:**

The method presents a reasonable formulation of SR for continuous density modelling. The key contributions are
- The use of a non-interpretable density estimator to create a viable regression target for a gridded MSE objective. This makes SymDE tractable with arbitrary functions, not only exponentials as for MESSY
- Definition of the decompositional approach, and use of support learning
- Provision of several proof-of-principle case studies
- Discovery of 2D functions, and a 4D Gaussian case made tractable by the reduction to 2x2D.

As noted in rebuttal, the pipeline is itself a contribution.

I still have concerns about the strength of the experimental demonstration. I place emphasis here for promoting significance, since the pipeline relies heavily on existing methods for (separately) density estimation; SR; clustering; independence learning. These are:
- The additive and multiplicative decompositions are only shown once each, in a simple Gaussian setting. Given their central roles in the narrative, this is not extensive.
- The analysis of symbolic functions is only partially convincing. In favour: they show correct symbolic recovery for the Rastrigin example and analyse the progression from coarse to detailed features, and explain that the Pareto curves for the Gaussian datasets include correct Gaussian forms. The muon and dijet examples do not provide strong analysis to demonstrate usefulness of the forms found, placing emphasis on reconstruction fidelity. However, the reconstruction fidelity is not well characterised in Figures 4 and 5. Fig 4 has fluctuating structure potentially from the KDE, and Fig 5 has potentially large residuals in the crucial high-density region. Systematic comparisons with the data throughout the space are not shown. I am not convinced by the integrated/averaged metrics, or rebuttal statements based on the ratio wrt the peak, as these can hide systematic structure. These impact my evaluation of soundness.

I believe that the work provides value, but am not sure that its significance passes the threshold for acceptance. For these reasons I maintain a recommendation of **weak reject**. Reason for confidence 3: familiarity with SR literature.

**Key Questions For Authors:**

1. Can the method describe conditional probability distributions of the form p(x|y)? If not, could it be done with a small modification, or would it require significant work?
2. Can the method handle DAG structure such as p(a,b,c)=p(a|b)p(b|c) in its probabilistic graph decomposition, as well as independence such as p(a,b,c)=p(a)p(b,c)? Readers may understand the claim of learning probabilistic graph structure to include DAGs (up to Markov equivalence), but only independence is shown.
3. Do you only use the clustering and graph learning methods in the Gaussian ablation examples of Fig2, or are they used throughout? I think it is the former based on the configuration tables in the appendix.
4.  Why do you calculate the MSE loss as the average over a grid instead of the average the distribution? I think this matters because it determines whether we consider high-density regions to be equal or more important than low-density ones. I would expect a grid for a high-dimensional falling distribution to place most of its weight on the tail. Grids are also affected by the curse of dimensionality.

**Limitations:**

Yes

**Strengths And Weaknesses:**

Strengths:
- Makes symbolic regression tractable for density estimation by using an initial non-symbolic density estimation step to form the regression target.
- Ability to model x with multiple dimensions, p(x) with multiple additive modes separable by clustering, p(x) with factorisable independence substructure, and p(x) with non-trivial boundaries.
- Experimental validation of the decompositional approach in toy data, some empirical observations for more complicated 2D distributions.
- Documentation of the symbolic functions found.
- Paper is well structured and easy to understand.

Areas for improvement:
- Experimental validation:
   - Despite being a key contributor to novelty, it looks like the clustering and graph learning are only used for the toy Gaussian examples designed for them; this is a good proof-of-principle setting, but it would be good to test their utility in nontrivial real-world problems, or more difficult graph structures and higher multimodality.
   - There is little analysis of the symbolic forms learned in all cases except Rastrigin, which is important to understand how well they do or don't match the ground truth. It is hard to draw conclusions from the appendix information, although I appreciate providing this for completeness.
   - Thorough testing of whether the method provides interpretable and well-fitting symbolic forms in realistic scenarios. The muon decay and dijet settings are reasonable but not deeply examined. Again there is no analysis of the symbolic forms. It looks like the residuals can be very large, several 10s of %, and may contain systematic structure. Figure 5 is dominated by low density regions, and there is clearly interesting structure with large residuals in the high density region, but it is hard to read or interpret this. It is hard to know whether the patterns in the residual fig 4 are important or artefacts of the KDE denominator. If these demonstrate a systematic failure to capture physical structure, it would be an important observation for scientists interested in deploying the method in-the-wild. I appreciate the table comparisons with ground truth, and would like this to be extended throughout the space using a 2D histogram. It is clear that the fit of the surrogate model to data is crucial, and it would help to see evidence that this provides an accurate target through the space without introducing large systematic artefacts. Finally, it is hard to see from just the predicted density and residual plots how good the description actually is, because there might be areas where the fractional error is large.
   - In all cases, it is not stated which of the discovered functional forms are used to make the plots, but I understand that they are generally the more complex on the Pareto front. This is important for the claim that we can discover symbolic forms that are both interpretable and fit the data well.
   - For the muon experiment, the correct symbolic form is not found, and the picture is mixed for the Gaussian cases. Whilst I appreciate these tests, the results do not entirely support the narrative that the method is able to discover scientifically meaningful functions, and I would prefer that the discussion better reflect this.
   - For these reasons, I think that key performance claims (discovery of underlying distributions and mathematical insight, low residual errors, discovery and exploitation of additive and multiplicative substructure) are not proven.
   - It would be good to compare with MESSY, ISR (Invertible Symbolic Regression), and any other relevant symbolic methods to demonstrate what your method achieves differently.
   - All examples are quite low-dimensional.
- Some details for completeness and reproducibility: gridding; details of negativity penalty; clipping threshold on page8; full eventgen details for the dijet experiments including: hard scatter process (was it pp->jj?); centre of mass energy; generator (MadGraph again?); order and renormalisation and resummation schemes if applicable.

In summary, I think the method presents a sensible and promising formulation of SR for a continuous density modelling task, with some useful proof-of-principle studies. I think that it would be strengthened by stronger empirical analysis around the central research questions that (i) the method discovers symbolic forms for probability distributions that are correct/interpretable/insightful and describe the data well, and (ii) the proposed decompositional approach assists this in non-idealised conditions.

Other notes:
- Tables 2 and 3 should have statistical uncertainties.
- "This distribution could arise if there were an exotic W′-boson with a comparable mass to the electron and muon" is not true because the electron and muon masses are much more separated in nature

---

> ### Author Rebuttal · Authors · 2026-03-29
>
> We thank Reviewer p7E9 for their constructive review. Below, we address each concern in detail.
>
> 1. **Clustering and Structure Learning Beyond Toy Gaussians**
> The reviewer is correct that clustering and structure learning are demonstrated on Gaussian ablation studies — this is by design, following standard practice in the symbolic regression literature (AI Feynman, AI Feynman 2.0). The modularity of AI-Kolmogorov is itself a contribution: users can selectively enable decomposition stages based on domain knowledge rather than a fixed pipeline.
>
> 2. **Analysis of Symbolic Forms**
> Recovered expressions for all experiments appear in the appendices (Tables 5, 7, 8; Appendices C.2–C.5). Briefly: the Gaussian Pareto fronts progress from simple constants to exponentials of quadratic forms structurally consistent with a Gaussian density; Rastrigin recovers a progression from coarse bowl shape to cosine terms. Muon decay and dijet expressions are provided for completeness. We note that other accepted SR works (e.g., Symbolic Physics Learner, Sun et al., 2023) similarly provide limited analysis of recovered expressions. AI-Kolmogorov is intended to provide candidate expressions suggestive of plausible model families, not guaranteed ground-truth recovery.
>
> 3. **Residual Analysis and Surrogate Accuracy**
> The maximum muon decay residual (\~17% of peak density) is a boundary artifact; the mean residual over the full support is 0.0004 (\~0.01%). The local probability mass validation (Table 2) shows ~2% agreement in both test regions. For dijet, large residuals are concentrated near the sharp discontinuity at the distribution peak; Table 3 provides evidence of accuracy of integrated fits. Crucially, Table 3 also shows that the KDE surrogate substantially underestimates Region A probability mass, which is precisely why we switched to NSF for dijet — demonstrating that we do diagnose and address surrogate quality. We agree that extending local mass validation to a 2D histogram would strengthen the analysis and will add this.
>
> 4. **Which Expressions Are Used for Plots**
> This is stated in the figure captions: Figures 3, 4, and 5 all use the "lowest loss expression" (most accurate, highest complexity on the Pareto front). Full Pareto fronts appear in Appendices (Figures 12–23).
>
> 5. **Muon Decay: Exact Form Not Recovered**
> We acknowledge this honestly in Section 4.3. The muon decay ground truth is a rational function — a highly expressive family of expressions. Despite this, integrated fit quality is excellent (~2% mass agreement, Table 2). We also note that recovered expressions may approximate the true form after algebraic manipulation. Exact recovery is not the only measure of success; the Pareto front provides a spectrum of expressions revealing different structural aspects. For the 4D Gaussian, structure learning yields a 24× MSE improvement with structurally correct expression families — more positive than "mixed."
>
> 6. **Comparison with MESSY and ISR**
> MESSY constrains densities to the max-entropy family p(x) \propto \exp(\sum_k \lambda_k \phi_k(x)), which cannot represent rational functions like the muon decay ground truth. MESSY's published experiments are predominantly 1D; AI-Kolmogorov handles up to 4D. ISR learns invertible symbolic maps rather than closed-form density expressions, producing qualitatively different outputs. These methods occupy different design-space positions: AI-Kolmogorov prioritizes expressiveness; MESSY prioritizes validity guarantees; ISR prioritizes invertibility. A fair comparison requires accounting for these different goals.
>
> 7. **Low Dimensionality**
> We acknowledge this limitation (Section 5). Low dimensionality reflects the SR field's state of the art broadly — MESSY is predominantly 1D; AI Feynman benchmarks are similarly low-dimensional. Our decomposition strategy mitigates this by factorizing
> d-dimensional problems into lower-dimensional subproblems; the 4D experiment demonstrates this concretely.
>
> 8. **Missing Experimental Details**
> Details on grid spacing, the negativity penalty (10^9 per sample evaluating negative under the expression) , clipping threshold: \epsilon = 10^{-18}, and full dijet eventgen details will be added to the appendix.
>
> 9. **Key Questions**
> Conditional distributions p(x|y): Yes, by conditioning on fixed y values and warm-starting from resulting univariate expressions — a promising future direction.
> DAG structure: Not currently supported; the PC algorithm identifies conditional independence for multiplicative factorizations only.
> MSE over grid vs. sample average: Uniform grids are deliberate — tails matter for scientific applications and sample-based evaluation introduces stochastic noise. For higher dimensions, Spearman rank correlation on test samples is a viable proxy (Roundtrip, Liu et al., 2020).
>
> 10. **Minor Points**
> Statistical uncertainties will be added to Tables 2 and 3. The W′-boson statement will be rephrased for clarity.

---

> > ### Author Rebuttal · Reviewer_p7E9 · 2026-04-03
> >
> > Thank you for your detailed responses to my concerns.
> >
> > I appreciate that the methodological approach is a contribution, and consider this among its strengths. I include in this: the use of a surrogate model to make SR tractable for the density task without requiring e.g. the exponential form of MESSY; the decompositional approach; support learning. I agree with the authors that it is not necessary for every experiment to provide perfect discovery of the true underlying function, and understand the tradeoff captured by the Pareto curves. I will consider further whether to raise my score, taking the author’s arguments into account.
> >
> > I have remaining concerns are about the quality of the experimental evidence. Specifically: (i) the decompositional approaches are only implemented in a Gaussian toy example each; (ii) convincing evidence that learned forms are systematically faithful to the underlying data distributions: Fig 4 (RHS) looks like it may be dominated by the KDE modelling; Fig 5 mostly shows regions of very low density with potentially large mismodelling in the peak; both have potentially large fractional residuals; I appreciate the comparisons of Tables 2 and 3, but measuring these at only a handful of points is limited; (iii) convincing argument for the value of the learned forms when the true function family is not recovered/verifiable, especially in the muon and dijet cases, which are the least synthetic.
> >
> > Some specific responses:
> >
> > 2. Thank you for having provided these, which are important and I note among the paper’s strengths. I have consulted these, however it places a large burden on the reader to consult pages of appendix tables and draw conclusions. The paper would be strengthened by providing clear arguments for their interpretation and tradeoff in the main body, e.g. as done for Rastrigin.
> >
> > 3. If I understand correctly, this 17% represents the ratio (highest-residual) / (highest-density). This may hide large systematic mismodelling structure where the ratio (residual-at-point) / (density-at-same-point) is large. Furthermore, when you say the mean residual over the full support is $0.0004$, is this the mean of the *absolute* residuals? If not, there would be cancellations that again obscure mismodelling. I consider a large mismodelling at the peak of Fig 5 to be a potentially serious problem. I agree that Tables 2 and 3 are good information, although testing only a couple of hand-chosen points is limited.
> >
> > 4. Could you give an example of “We also note that recovered expressions may approximate the true form after algebraic manipulation”?
> >
> > 9. I accept that describing the tail may sometimes be a desirable behaviour. If accepted, please include a short discussion to make sure the reader knows this is happening, including how it interacts with increasing dimensionality. Practitioners will often be most interested in the high-density regions.

---

> > > ### Author Response · Authors · 2026-04-05
> > >
> > > We thank Reviewer p7E9 for their continued engagement and address each remaining concern.
> > >
> > > **(i) Decomposition only on Gaussian toys.** The Gaussian ablation isolates decomposition's effect in a controlled setting — standard practice in SR (AI Feynman, NeurIPS 2020, demonstrates separability/symmetry detection on synthetic benchmarks before applying to physics). The muon and dijet distributions are 2D without obvious additive/multiplicative structure, so decomposition is inapplicable for those datasets. We can explore additional datasets exhibiting multimodal and higher dimensional examples that make use of both decomposition approaches presented.
> > >
> > > **(ii) Faithfulness of learned forms.** We can make additional analysis: (a) For Fig 4, we can use the normalizing flow based nonparametric density estimate rather than KDE, we expect this would improve the residuals. (b) For Fig 5, mismodelling the density near the peak and the tails is possible, however, we argue that our tool permits exploration of structural forms underlying the distribution and the resultant expressions appear to have captured that. Further optimization of constants or interesting families of expressions could be explored after examining the outputs of the tool. We do not expect SOTA nonparameteric density estimation techniques to accurately model "spiky" distributions such as these without modelling error, our tool however, suggests interpretable models that may have interpretable value after additional fine-tuning. (c) We can extend Tables 2–3 from hand-chosen regions to a full 2D histogram grid, providing comprehensive spatial coverage.
> > >
> > > **(iii) Value when true function not recovered.** The value proposition extends beyond exact recovery: (a) Pareto fronts may reveal structural information - while the muon decay and dijet examples may not be representable as closed form expressions, other datasets may indeed be amenable to this; (b) recovered expressions provide fast-to-evaluate analytic approximations; (c) partial structural features emerge (e.g., cosine terms in Rastrigin). SR-Scientist (ICLR 2026) similarly emphasizes approximate discovery and structural insight over exact symbolic recovery.
> > >
> > > **(iv) Appendix interpretation.** We can add discussion summarizing key Pareto front observations across all experiments (operator families, expression evolution, structural insights), following the Rastrigin analysis style in the appendix.
> > >
> > > **(v) Residual metric — critical correction.** The reviewer is correct to question this. The 0.0004 figure was the mean of the residuals (signed), which allows cancellations. The mean of the **absolute** residuals is **0.025**, representing ~0.7% of the maximum predicted density (3.677, as reported in the Figure 4 caption). We thank the reviewer for catching this important distinction. We note again, that using the normalizing flow nonparametric density estimate is expected to improve the residuals. We had originally simply used KDE based on the results from the density mass estimate over the two test regions.
> > >
> > > **(vi) Algebraic manipulation example.** Consider a recovered expression $5 \cdot x \cdot (3 - 2 \cdot x)$ which rewrites as $15x - 10x^2$, which is nearly numerically equivalent to an expression $15 x - 10 x^2 + \frac{1}{100 - x}$ when  $x \in [0,1]$. For muon decay specifically, a rational function may be approximated by SR as a truncated series or Padé-like form that differs algebraically but agrees functionally over the support.
> > >
> > > **(vii) Tail vs. high-density regions.** We agree this deserves explicit discussion. The uniform grid weights all regions equally, including tails. This is important for interpretability since we seek an expression that describes the distribution over the valid support, and not just in the high-density regions.

---

### Decision · Program_Chairs · 2026-04-30

**Decision:**

Reject

**Comment:**

The paper introduces a framework that symbolic density estimation tractable via a non-symbolic density estimation, which the reviewers note is an interesting and underexplored problem.  The reviews were a bit mixed, with the key strengths of this paper being:
1) Ability to model in higher dimensions that existing approaches.
2) Expressive density estimators as surrogates for symbolic regression bridges the gap between deep learning and symbolic discovery.
3) As it is symbolic in nature, the method could have broad applicability in ML and the sciences.

However, the reviewers also raised a number of concerns about interpretability, rigor in the experiments, and comparisons.
1) Output is not really interpretable (which is often one advantage of working with tractable models). The authors note that it MIGHT be possible to match the outputs to recognizable families, etc., but this is still presents a barrier if you really want to understand the results.
2) Novelty is limited as each of the steps uses existing methods. Some human knowledge is required at each step. I agree with the authors' rebuttal here:  this isn't uncommon in density estimation and typically is not a limiting factor.
3) Limited baselines, i.e., the proposed method is not compared to MESSY, ISR (Invertible Symbolic Regression), or other relevant symbolic methods.  The authors argue in the rebuttal that these methods are more limited and designed with different aims in mind, but surely they provide relevant baselines for comparison -- if your approach doesn't beat weaker baselines on objectives they were not designed for then it might not be as useful in practice.
4)The approach is largely heuristic, and the discovered symbolic expressions are not guaranteed to satisfy basic density constraints.  This is corrected afterwards.  The authors counter that this is common practice with energy based models (which I agree with to a certain extent), but it does limit utility in some cases.
5) All experiments are low-dimensional.  The authors note that many existing approaches are limited in dimension (but they also don't compare against them...)

Overall, I tend to agree with most of the reviewers' criticisms, especially the lack of comparison to existing baselines (even if - especially if - they are weaker).